# In vitro reconstitution of an efficient nucleotide excision repair system using mesophilic enzymes from *Deinococcus radiodurans*

Anna Seck [1,2], Salvatore De Bonis[1], Christine Saint-Pierre[2], Didier Gasparutto [2], Jean-Luc Ravanat [2✉] & Joanna Timmins [1✉]

Nucleotide excision repair (NER) is a universal and versatile DNA repair pathway, capable of removing a very wide range of lesions, including UV-induced pyrimidine dimers and bulky adducts. In bacteria, NER involves the sequential action of the UvrA, UvrB and UvrC proteins to release a short 12- or 13-nucleotide DNA fragment containing the damaged site. Although bacterial NER has been the focus of numerous studies over the past 40 years, a number of key questions remain unanswered regarding the mechanisms underlying DNA damage recognition by UvrA, the handoff to UvrB and the site-specific incision by UvrC. In the present study, we have successfully reconstituted in vitro a robust NER system using the UvrABC proteins from the radiation resistant bacterium, *Deinococcus radiodurans*. We have investigated the influence of various parameters, including temperature, salt, protein and ATP concentrations, protein purity and metal cations, on the dual incision by UvrABC, so as to find the optimal conditions for the efficient release of the short lesion-containing oligonucleotide. This newly developed assay relying on the use of an original, doubly-labelled DNA substrate has allowed us to probe the kinetics of repair on different DNA substrates and to determine the order and precise sites of incisions on the 5′ and 3′ sides of the lesion. This new assay thus constitutes a valuable tool to further decipher the NER pathway in bacteria.

[1] Univ. Grenoble Alpes, CEA, CNRS, IBS, F-38000 Grenoble, France. [2] Univ. Grenoble Alpes, CEA, CNRS, SyMMES-UMR 5819, F-38000 Grenoble, France. ✉email: jean-luc.ravanat@cea.fr; Joanna.timmins@ibs.fr

Nucleotide-excision repair (NER) is one of several DNA-repair pathways that are universal. NER is a versatile pathway[1–4], capable of removing a very wide range of chemically and structurally diverse lesions, including adducts caused by smoking or generated by chemotherapy and UV-induced lesions such as pyrimidine–pyrimidone (6–4) photo-products (6-4-PP) and cyclobutane pyrimidine dimers (CPD). The common feature of this diverse set of DNA lesions is believed to be their ability to distort or destabilize the DNA duplex. However, defining what constitutes a helix-distorting lesion is not straightforward, and at present, the substrate specificity of the NER pathway remains poorly defined. Ribonucleotides that are known to cause only weak distortions to the DNA duplex (B- to A-form transition) have, for instance, been found to be substrates for the NER system[5].

Bacterial NER is mediated by the sequential action of six proteins: the four UvrA, UvrB, UvrC, and UvrD proteins, the DNA polymerase I, and DNA ligase[1–4]. UvrA, acting as a dimer, together with UvrB, is responsible for DNA-damage recognition[6–8]. After damage recognition, UvrA dissociates from the DNA, while UvrB forms a stable pre-incision complex upon sites of DNA damage and recruits UvrC[1–4]. The detailed molecular mechanisms underlying this recruitment step are still only poorly understood. UvrC is an enzyme possessing a dual endonuclease activity: one located at its N-terminus that is responsible for the 3′ incision and another located at its C-terminus that is in charge of the 5′ incision[1–4]. The dual incision of the damage-containing strand by UvrC yields a 12- or 13-nucleotide fragment containing the damaged base, which is released from the DNA duplex by the DNA helicase, UvrD[1–4]. The gap is subsequently filled and sealed by the combined actions of the DNA polymerase I and DNA ligase.

Bacterial NER has been reconstituted in vitro using the three essential proteins, UvrA, UvrB, and UvrC, and either plasmid or short DNA oligonucleotides as substrates[9–13]. Although several of the early studies made use of Escherichia coli Uvr proteins[9,12–21], many of the more recent mechanistic studies of bacterial NER have relied on the use of Uvr proteins from thermophilic bacteria (Bacillus caldotenax, Geobacillus stearothermophilus, Thermatoga maritima, and Thermus thermophilus)[6,10,22–27] and, in many cases, due to solubility issues, Uvr proteins from different sources were combined to set up functional incision assays.

Deinococcus radiodurans is a nonpathogenic, mesophilic bacterium, which displays an exceptional ability to withstand the lethal effects of DNA-damaging agents, including ionizing radiation and UV light. D. radiodurans can survive without loss-of-viability UV doses up to 500 J/m$^2$ [28,29], a dose that is known to generate ~5000 pyrimidine dimers per genome copy. A number of factors, including a high intracellular concentration of anti-oxidant metabolites, a well-protected proteome, an efficient DNA repair machinery, and a high copy number of its genome, have been proposed to contribute to this robust radiation-resistant phenotype[30]. The genome of D. radiodurans encodes for a complete NER pathway[31]: UvrA1 (DR_1771), UvrB (DR_2275), UvrC (DR_1354), and UvrD (DR_1775), but also for a UV-damage endonuclease, UvsE, that efficiently repairs UV-induced pyrimidine dimers[32]. In a uvrA1 knock out strain of D. radiodurans, UvsE can compensate in part for the absence of UvrA1[33]. In addition, a gene encoding for a second, class-II UvrA protein (UvrA2, DR_A0188) can be found[31]. Based on transcriptomics data, however, under normal growth conditions, UvrA2 is approximately ten times less abundant in D. radiodurans cells than its highly conserved counterpart, UvrA1[34]. The expression of all five uvr genes, but not the uvsE gene, is upregulated 3–5-fold following exposure to ionizing radiation[34]. UvrA2 proteins are found in many bacteria living in harsh environments and show a high degree of sequence similarity to UvrA1, but are missing the proposed UvrB-interaction domain (Supplementary Fig. 1)[35]. Despite this deletion, there is evidence that UvrA2 may play a minor role in DNA repair and tolerance to DNA-damaging agents, including UV, but it is unclear whether these UvrA variants are directly implicated in NER[33,35]. UvrA2 has also been proposed to take part in export of damaged oligonucleotides, a process that is known to occur in irradiated D. radiodurans[31].

In the present study, we have established a robust incision assay relying on the activity of highly pure UvrA1, UvrB, and UvrC from a single, mesophilic organism, the radiation-resistant bacterium, D. radiodurans[31]. In contrast to earlier work, this assay makes use of a doubly-labeled DNA substrate allowing us to specifically identify the different incision products. Moreover, this assay has enabled us to assess the possible involvement of UvrA2 in NER, to perform kinetics studies of the NER process, to probe the substrate specificity of D. radiodurans NER, and to evaluate the role of divalent cations in the dual-incision reaction. Finally, by combining this NER assay with MALDI-ToF mass spectrometry analyses, we unambiguously determined the sites of cleavage by UvrC.

## Results

**Reconstitution of a functional NER system in vitro.** Individual recombinant Uvr proteins, drUvrA1, drUvrA2, drUvrB and drUvrC, from Deinococcus radiodurans, were expressed in and purified from E. coli (Fig. 1a). For each subunit, at least three chromatographic steps were used to ensure that the protein samples were of high purity and devoid of any nucleic acid contamination. Incision reactions were performed using a 50 mer duplex in which one strand was 5′ end-labeled with a red fluorophore and contained a fluorescein-conjugated thymine (FdT) in position 26 (F26-seq1), a well-established substrate of bacterial NER[3,36] (Fig. 1b). Dual incision of this substrate produces a 12 mer fragment bearing only the green fluorophore, an 18 mer fragment bearing the red fluorophore, and a third 20 mer fragment with no fluorophores that could therefore not be seen on TBE-urea gels (Fig. 1b, d). In addition, intermediate fragments resulting from either 5′ incision (32 mer fragment) or 3′ incision (30 mer fragment) only could be seen under certain experimental conditions, as described below. The 32 mer fragment resulting from 5′ incision bears only the green fluorophore, while the 30 mer fragment resulting from 3′ incision bears both the green and red fluorophores (Fig. 1b, d). So, although these two bands were difficult to distinguish based on their migration, their different fluorescent properties allowed unambiguous interpretation of the products of the reactions.

Once the purification protocols for the three Uvr proteins, and notably for UvrC (Supplementary Fig. 2 and "Materials and methods"), were optimal, different combinations of Uvr proteins were tested. Efficient dual incision was only observed when drUvrA1, drUvrB, and drUvrC proteins were included in the reaction (Fig. 1c). No cleavage was observed when using a substrate with just the 5′ end-labeled red fluorophore and missing the FdT moiety (Supplementary Fig. 2). Different concentrations of each of the Uvr proteins were tested to find the optimal conditions (Supplementary Fig. 2c), which led to ~80% incision of the 50 mer duplex after 1 hour incubation at 37 °C (Fig. 1c). The incision activity was largely independent of drUvrA1 concentration, whereas changing drUvrB or drUvrC concentrations clearly affected the incision efficiency (Supplementary Fig. 2c). The final conditions were set to 1 μM drUvrA1, 0.5 μM drUvrB and 2 μM drUvrC, and were performed using 25 nM DNA substrate. Under these conditions, the class II variant of UvrA, drUvrA2, was not functional and could not replace drUvrA1 in the incision reaction (Fig. 1d). Addition of up to an

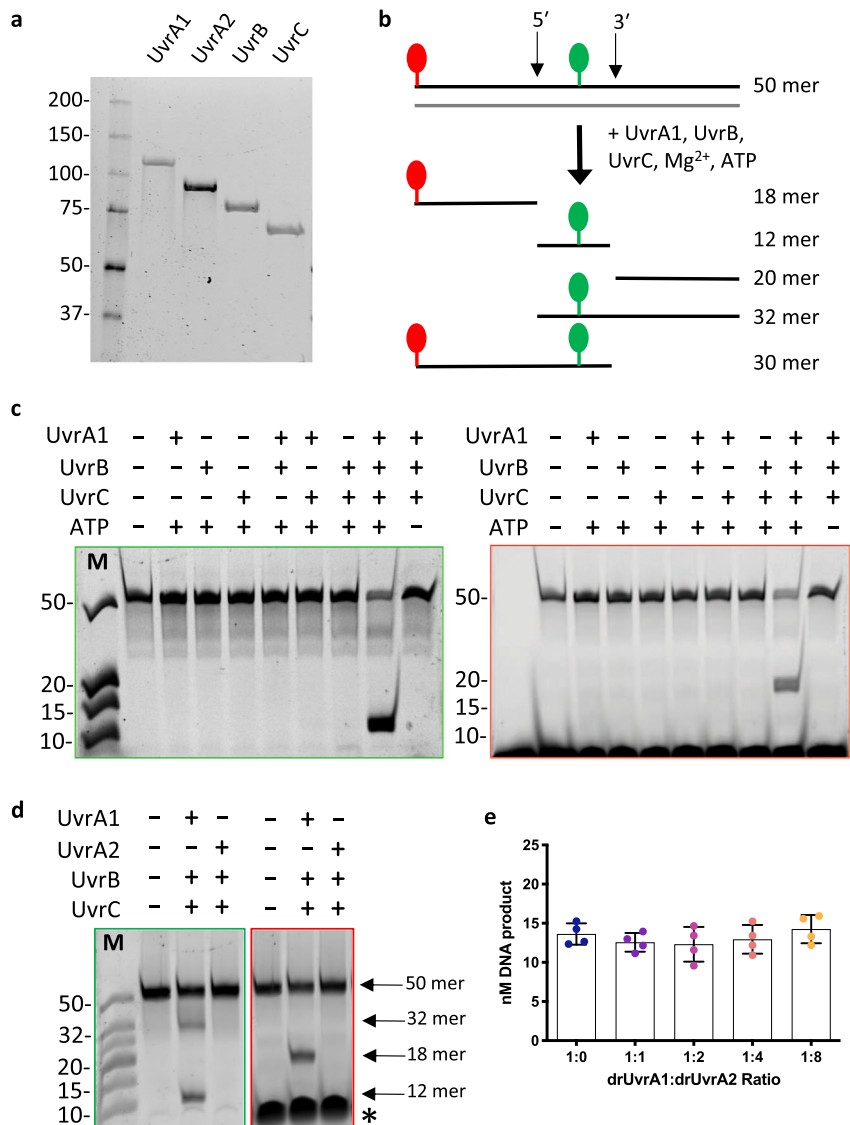

**Fig. 1 Deinococcus radiodurans UvrABC system. a** SDS-PAGE analysis of the purified drUvrA1, drUvrA2, drUvrB, and drUvrC proteins from *D. radiodurans*. The first lane corresponds to molecular weight markers in kDa. **b** Schematic diagram illustrating the design of the dsDNA substrates used in this study and the sites of incision by drUvrABC on the 5′ and 3′ sides of the lesion, corresponding to a fluorescein-conjugated thymine (green). An additional red fluorophore was added to the 5′ end of the substrates to allow to differentiate the DNA fragments released on the 5′ and 3′ sides of the lesion. **c** TBE-polyacrylamide urea-gel analysis of the drUvrABC incision activity in the presence or absence of each of the three Uvr proteins or ATP. Reactions were performed for 1 hour at 37 °C using 25 nM F26-seq1 substrate and different combinations of drUvrA1 (1 μM), drUvrB (0.5 μM), and drUvrC (2 μM) in the presence of 2.5 mM Mg$^{2+}$ and 2.5 mM ATP. **d** TBE-polyacrylamide urea-gel analysis of the drUvrABC incision activity in reactions containing either 1 μM drUvrA1 or 1 μM drUvrA2. Reaction conditions were the same as in (**c**). The major bands observed by electrophoresis using either the green- or red filter are indicated with arrows. The large band indicated with a * in the red channel corresponds to the sample-loading dye that produces a strong fluorescence in the red filter. **c–d** Green-boxed gels were visualized with the green filter to detect fluorescein-labeled bands, whereas red-boxed gels were visualized with the red filter to detect ATTO633-labeled bands. Left lane: molecular weight marker composed of fluorescein-labeled oligonucleotides ranging from 10 to 50 bp. **e** Effect of drUvrA2 on the incision reaction performed by drUvrABC. Reactions were performed at 37 °C for 45 min using 25 nM F26-seq1 substrate, 0.25 μM drUvrA1, 0.5 μM drUvrB and 2 μM drUvrC (blue), supplemented with 0 (ratio 1:0), 0.25 (ratio 1:1), 0.5 (ratio 1:2), 1 (ratio 1:4), or 2 μM (ratio 1:8) drUvrA2. All reactions contained 2.5 mM MgCl$_2$ and were started by addition of 2.5 mM ATP. Dot-plots present the mean amount of 12 mer product (nM) released and standard deviation of four individual replicates illustrated as individual dots.

8-fold excess of drUvrA2 to the drUvrABC reaction also did not influence the incision activity, suggesting that drUvrA2 is not directly involved in NER in *D. radiodurans* (Fig. 1e).

ATP and magnesium (Mg$^{2+}$) were also found to be essential cofactors of the incision reaction and the highest incision efficiency was obtained when they were added at a 1:1 ratio at a final concentration of 2.5 mM (Fig. 2a, b). The ATP could not be replaced by ADP or nonhydrolyzable analogs of ATP

(Supplementary Fig. 3a), suggesting that both ATP binding and hydrolysis were necessary for efficient incision by drUvrABC. The ATP was thus used to start the reaction after a 5 min preincubation of the drUvrABC system with the DNA substrate. Alternatively, similar results were obtained by starting the incision reaction by addition of drUvrC after a 10 min preincubation of drUvrA and drUvrB with ATP and the DNA substrate (Supplementary Fig. 3b).

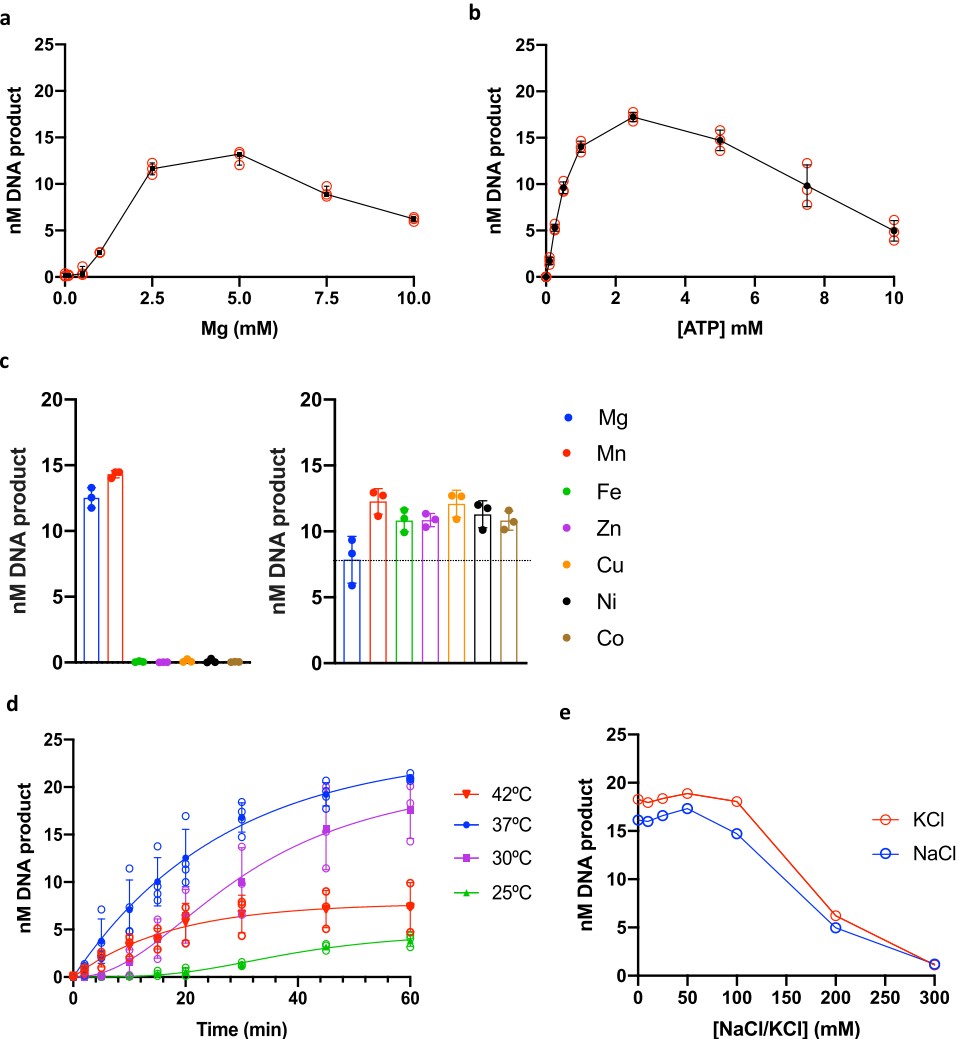

**Fig. 2 Optimization of the in vitro NER system. a** Dual-incision activity by drUvrABC as a function of MgCl$_2$ concentration. Reactions were performed at 37 °C using 25 nM F26-seq1 substrate, 0.25 µM drUvrA1, 0.5 µM drUvrB and 2 µM drUvrC supplemented with 0, 0.1, 0.25, 0.5, 1.0, 2.5, 5.0, 7.5, or 10.0 mM MgCl$_2$. Reactions were started by addition of 2.5 mM ATP. The graph presents the mean amount of 12 mer product (nM) released after 30 minutes (black symbols) and the standard deviation of three individual replicates shown as open red circles. **b** Dual-incision activity by drUvrABC as a function of ATP concentration. Reactions were performed at 37 °C using 25 nM F26-seq1 substrate, 1 µM drUvrA1, 0.5 µM drUvrB and 2 µM drUvrC, and 2.5 mM MgCl$_2$. Reactions were started by addition of 0, 0.1, 0.25, 0.5, 1.0, 2.5, 5.0, 7.5 or 10.0 mM ATP. The graph presents the mean amount of 12 mer product (nM) released after 45 minutes (black symbols) and the standard deviation of three individual replicates shown as open red circles. **c** Effects of metals on the incision activity by drUvrABC. Left: Dual incision activity in the presence of 2.5 mM of Mg (blue), Mn (red), Fe (green), Zn (purple), Cu (orange), Ni (black), or Co (brown). Reactions were performed at 37 °C using 25 nM F26-seq1 substrate, 1 µM drUvrA1, 0.5 µM drUvrB and 2 µM drUvrC. Reactions were started by addition of 2.5 mM ATP. The dot plots present the mean amount of 12 mer product (nM) released after 30 minutes and standard deviation of three individual replicates shown as filled circles. Right: Dual incision activity in the presence of 2.5 mM Mg alone (blue), or 2.5 mM Mg supplemented with 0.25 mM of Mn (red), Fe (green) Zn (purple), Cu (orange), Ni (black), or Co (brown). Reactions were performed at 37 °C using 25 nM F26-seq1 substrate, 1 µM drUvrA1, 0.5 µM drUvrB and 2 µM drUvrC. Reactions were started by addition of 2.5 mM ATP. The dot plots present the mean amount of 12 mer product (nM) released after 20 minutes and standard deviation of three individual replicates shown as filled circles. The dashed line indicates the extent of incision in the presence of Mg alone. **d** Effects of temperature on the drUvrABC incision activity. Time-course experiments were performed at 25 (green), 30 (purple), 37 (blue), and 42 °C (red) for 1 hour. The graph presents the mean amount of 12 mer product (nM) released at each timepoint (filled triangles) and standard deviation of at least three individual replicates shown as open circles. **e** Effects of salt on the drUvrABC incision activity. Dual incision activity by drUvrABC as a function of NaCl (blue) and KCl (red) concentration. Reactions were performed at 37 °C using 25 nM F26-seq1 substrate, 1 µM drUvrA1, 0.5 µM drUvrB and 2 µM drUvrC in reaction buffer containing 0, 10, 25, 50, 100, 200, or 300 mM NaCl or KCl. Reactions were started by addition of 2.5 mM ATP. The graph presents the amount of 12 mer product (nM) released after 45 minutes of reaction.

We also investigated whether other divalent or trivalent metal cations ($Fe^{3+}$, $Co^{2+}$, $Mn^{2+}$, $Ni^{2+}$, $Zn^{2+}$, and $Cu^{2+}$) were needed for this reaction either in replacement of or in addition to $Mg^{2+}$ (Fig. 2c). Of these, only $Mn^{2+}$ could efficiently replace $Mg^{2+}$. No incision activity could be detected when $Fe^{3+}$, $Co^{2+}$, $Zn^{2+}$, $Ni^{2+}$, or $Cu^{2+}$ were used instead of $Mg^{2+}$. In contrast, when these metals were mixed at a 1:10 molar ratio with $Mg^{2+}$, they all mildly enhanced the incision activity (Fig. 2c). Interestingly, when $Mg^{2+}$ was substituted with $Mn^{2+}$, nonspecific bands were detected on the gels in the absence of drUvrA1, suggesting that $Mn^{2+}$ may favor nonspecific incision activity (Supplementary Fig. 4a). This spurious activity could be blocked by addition of

**Table 1 Expected and measured masses of DNA fragments after processing of F26-seq1 by drUvrABC.**

| Oligonucleotide | Sequence | Expected mass (Da) | Measured mass (Da) |
|---|---|---|---|
| 5′-ATTO633-F26-seq1 (substrate) | 5′- **X**GAC TAC GTA CTG TTA CGG CTC CAT C**FdT**C TAC CGC AAT CAG GCC AGA TCT GC -3′ | **16,465.0** | **16,457.5** |
| Rev-seq1 (substrate) | 5′- GCA GAT CTG GCC TGA TTG CGG TAG AGA TGG AGC CGT AAC AGT ACG TAG TC -3′ | **15,531.0** | **15,529.7** |
| 12mer (product) | 5′- **p**CTCCATC**FdT**CTAC -3′ | **4,107.4** | **4,109.1** |
| 18mer (product) | 5′- **X**GACTACGTACTGTTACGG -3′ | **6,226.6** | **6,226.5** |
| 20mer (product) | 5′- **p**CGCAATCAGGCCAGATCTGC -3′ | **6,167.0** | **6,165.4** |

*X* ATTO633 moiety, *FdT* fluorescein-conjugated thymine, *p* phosphate.

$0.25 \, mM$ $Fe^{3+}$ to the reactions containing $2.5 \, mM$ $Mn^{2+}$ (Supplementary Fig. 4b). The incision activity of drUvrABC is thus sensitive to the abundance of metal cations.

To further optimize the reaction conditions, we also varied the temperature at which the reaction was performed (25, 30, 37, and 42 °C) and the salt concentration (NaCl and KCl) in the reaction buffer (Fig. 2d, e). The strongest incision activity was obtained at 37 °C with 50 mM KCl in the reaction buffer. At 37 °C, the incision activity was robust, long-lasting, and followed a single exponential model, whereas at lower temperatures, the reactions were much slower with the appearance of a clear lag phase at the start of the reaction, causing the curves to adopt a sigmoidal shape. The lag phase was particularly long at 25 °C, where no incision was observed during the first 15 minutes of the reaction. Interestingly, no such lag phase was seen at 42 °C, but the reaction reached a plateau after 30 minutes that was much lower than that observed at 37 °C with only 20% of DNA incision, indicating that the Uvr subunits may not be very stable at this temperature (Fig. 2d). As for the salt concentration, it was clear that concentrations of NaCl or KCl above 100 mM led to a major drop in the dual-incision activity of UvrABC (Fig. 2e).

**Identification of the cleavage product.** Bacterial NER has been reported to produce a 12 or 13 mer fragment, depending on the substrate[3,17]. To determine the precise sites of incision on the 5′ and 3′ sides of the lesion and the exact size of the fragments resulting from the incisions by the *D. radiodurans* NER system, we analyzed the products of the incision reaction performed on F26-seq1 DNA substrate by MALDI-ToF mass spectrometry (Table 1 and Fig. 3). The F26-seq1 substrate is composed of the oligonucleotides 5′-ATTO633-F26-seq1 and Rev-seq1 with the respective theoretical masses of 16,465.0 Da and 15,531.0 Da. Accordingly, two peaks at 16,457.5 Da and 15,529.7 Da were detected in the MALDI-ToF mass spectra of the substrate (Table 1 and Fig. 3a). After processing by drUvrABC, several additional DNA fragments were detected on the MALDI-ToF mass spectra. The final fragment of the dual incision containing the FdT was detected at 4109.1 Da corresponding to a 12 mer oligonucleotide with a phosphate at its 5′ end resulting from cleavage of the DNA 7 nucleotides upstream of the lesion and 4 nucleotides downstream of the lesion (Table 1 and Fig. 3b). In agreement with this, an 18 mer fragment bearing the red fluorophore at its 5′ end resulting from 5′ incision was detected at 6226.5 Da, close to its theoretical mass of 6226.6 Da, and a 20 mer fragment carrying a phosphate at its 5′ end resulting from 3′ incision was also detected at 6165.4 Da, close to its theoretical mass of 6167.0 Da.

**Cleavage order and single-turnover repair kinetics.** After optimization of the reaction conditions, the extent of incision reached ~80% after 1 hour with almost no accumulation of intermediate fragments. Figure 4a presents a typical single-turnover time-course reaction in which F26-seq1 was treated with an excess of drUvrABC system. To ensure we were indeed in single-turnover reaction conditions, we estimated the first-order observed rate constants ($k_{obs}$) of the reaction at lower UvrABC concentrations, but saw no difference when lowering the concentrations 2-, 4-, or 10-fold (Supplementary Table 1). We therefore decided to pursue single-turnover experiments using 1 µM drUvrA1, 0.5 µM drUvrB and 2 µM drUvrC. The amount of substrate decreased from 25 nM at the start to less than 5 nM after 1 hour of reaction, and conversely the 12 mer product accumulated to near 20 nM after 1 hour. The disappearance of the substrate results from a single-incision reaction on either the 5′ or 3′ side of the lesion, whereas the accumulation of 12 mer results from the sequential dual incision reaction and its rate is thus defined by the rate-limiting step, i.e., the slowest of the two reactions. In contrast, the accumulation of the intermediate 18 mer fragment results solely from the 5′ incision reaction. Interestingly, we observed that the kinetics of production of the 12 mer and 18 mer fragments were very similar ($k_{obs} \approx k_{obs1}$), both following a single exponential with observed rates of $0.031 \pm 0.003 \, min^{-1}$ and $0.030 \pm 0.010 \, min^{-1}$ respectively (Fig. 4a). This indicates that the second cleavage reaction resulting in the release of the 12 mer product occurs immediately after the first incision of the lesion ($k_{obs2} \gg k_{obs1}$) and that the first incision reaction is thus the rate-limiting step in the dual incision reaction. The two cleavage reactions are thus quasi-simultaneous. Interestingly, the observed rate of decay of the 50 mer substrate, estimated to be $0.040 \pm 0.011 \, min^{-1}$, is a little higher than the observed 5′ incision rate, indicating that this decay is not only due to the 5′ incision of the 50 mer, but also to some 3′ incision, indicating that the first incision reaction can occur on either the 5′ or 3′ side of the lesion. In these time-course measurements, we noticed that the amount of 18 mer fragment, resulting from cleavage on the 5′ side of the FdT, plateaued at a lower amount than for the 12 mer fragment resulting from the dual incision reaction (Fig. 4a). This may be due to further processing of the 18 mer fragment into smaller DNA fragments, which were not resolved in our gels.

While optimizing the $Mg^{2+}$ concentration needed for efficient incision by drUvrABC, we observed that $Mg^{2+}$ clearly modulated the overall incision efficiency, but also differentially regulated the 5′ and 3′ cleavage activities by UvrC (Fig. 4b). As discussed above, using our optimal conditions at 2.5 mM $Mg^{2+}$, 5′ and 3′ cleavage reactions occurred very efficiently and quasi-simultaneously, whereas at either lower (1 mM) or higher (10 mM) concentrations, the amounts of 12 mer product were severely reduced (Figs. 2a, 4b). Time-course experiments at these three $Mg^{2+}$ concentrations were performed to allow a more thorough comparison of these different reactions (Fig. 4b and Supplementary Fig. 5). Interestingly, we observed that at 1 mM $Mg^{2+}$, the 12

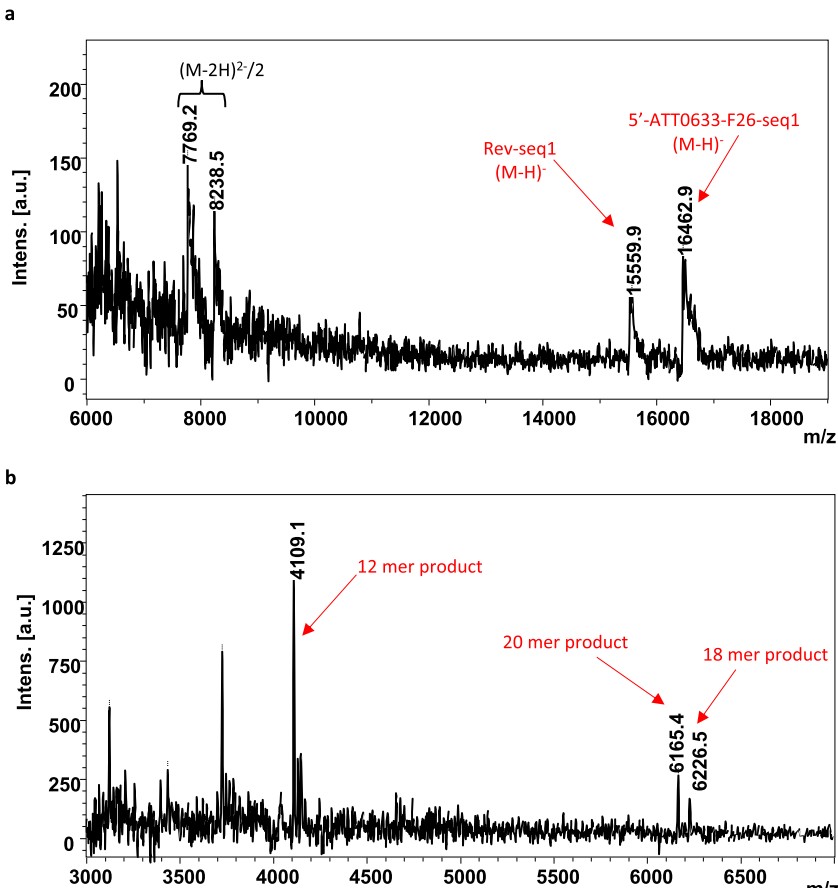

**Fig. 3 MALDI-ToF mass spectra of the drUvrABC incision-reaction substrates and products.** The incision reactions were performed at 37 °C for 1 hour using 25 nM F26-seq1 substrate, 1 µM drUvrA1, 0.5 µM drUvrB and 2 µM drUvrC, 2.5 mM MgCl$_2$, and 2.5 mM ATP. Peaks corresponding to either substrates (**a**) or products (**b**) of the incision reaction are indicated with red arrows. Masses of major peaks are indicated in Da. In (**a**), the two peaks with m/z values close to 8000 correspond to the doubly charged forms of the starting oligonucleotides bearing the lesion (5′-ATTO633-F26-seq1) and its complementary strand (Rev-seq1).

mer fragment production dropped, indicating that at this Mg$^{2+}$ concentration, the rate of incision is severely reduced. The very low amounts of intermediate products (in this case, the 30 mer fragment) seen to accumulate under these conditions suggest that the cleavage reactions are nonetheless still quasi-simultaneous as at 2.5 mM MgCl$_2$. This indicates that both the 5′ and 3′ incision reactions are impaired at low magnesium concentration. In contrast, at 10 mM Mg$^{2+}$, we observed a marked accumulation of the intermediate 32 mer fragment caused by a reduced 3′ incision activity. Mg$^{2+}$ thus plays a very critical role in fine-tuning the dual-incision activity of drUvrABC.

**Substrate specificity**. We next examined the substrate specificity of the *D. radiodurans* NER system by evaluating its repair efficiency on three additional substrates, all of which were 50 mer DNA duplexes containing at least one fluorophore for detection and a modified base in position 26 (Supplementary Tables 2, 3). As mentioned above, we first verified that an unmodified 5′-ATTO633-labeled 50 mer duplex was not a substrate of *D. radiodurans* NER (Supplementary Fig. 2a). A second FdT-containing substrate was prepared, F26-seq2, which only differs from F26-seq1 in terms of sequence. The bases on either side of the FdT were randomly changed to create a new DNA sequence with the same overall GC content as F26-seq1. A third substrate, named B26-seq1, was prepared, which shares the same sequence as F26-seq1, but bears a biotin-conjugated thymine (BdT) in

position 26 instead of the FdT. Finally, a fourth, bulkier NER substrate was prepared by binding streptavidin to the B26-seq1 substrate to form B26-seq1-strep. As with F26-seq1, all three of these substrates were 5′ end-labeled with a red fluorophore (ATTO633). Single-turnover kinetic measurements were performed on all four substrates (Fig. 5a, b). The kinetics of repair of the two FdT-containing substrates both followed a simple exponential, but the observed rate of release of the 12 mer fragment from the F26-seq2 substrate was considerably faster than with F26-seq1, with a K$_{obs}$ of 0.074 ± 0.007 min$^{-1}$ versus 0.031 ± 0.003 min$^{-1}$ for F26-seq1. This suggests that F26-seq2 is a better substrate than F26-seq1 (Fig. 5a). For the two biotin-containing substrates, B26-seq1 and B26-seq1-strep, the kinetics of repair were clearly altered with the time courses displaying a major delay at the start of the reaction causing the curves to adopt a sigmoidal shape in contrast to the simple exponential curves obtained for FdT substrates (Fig. 5a, b). After this initial delay, the observed rates of 12 mer release were similar to those obtained with F26-seq1, with a K$_{obs}$ of 0.034 ± 0.002 min$^{-1}$ for B26-seq1 and 0.037 ± 0.003 min$^{-1}$ for B26-seq1-strep (Supplementary Table 1). However, unlike with F26-seq1, we observed that the rates of product release were much lower with B26-seq1 when lower concentrations of drUvrABC were used (Supplementary Table 1), indicating that BdT is a poorer NER substrate than FdT.

To determine the possible cause of the delay at the start of the incision reaction observed with BdT-containing substrates, we performed fluorescence-polarization measurements to establish

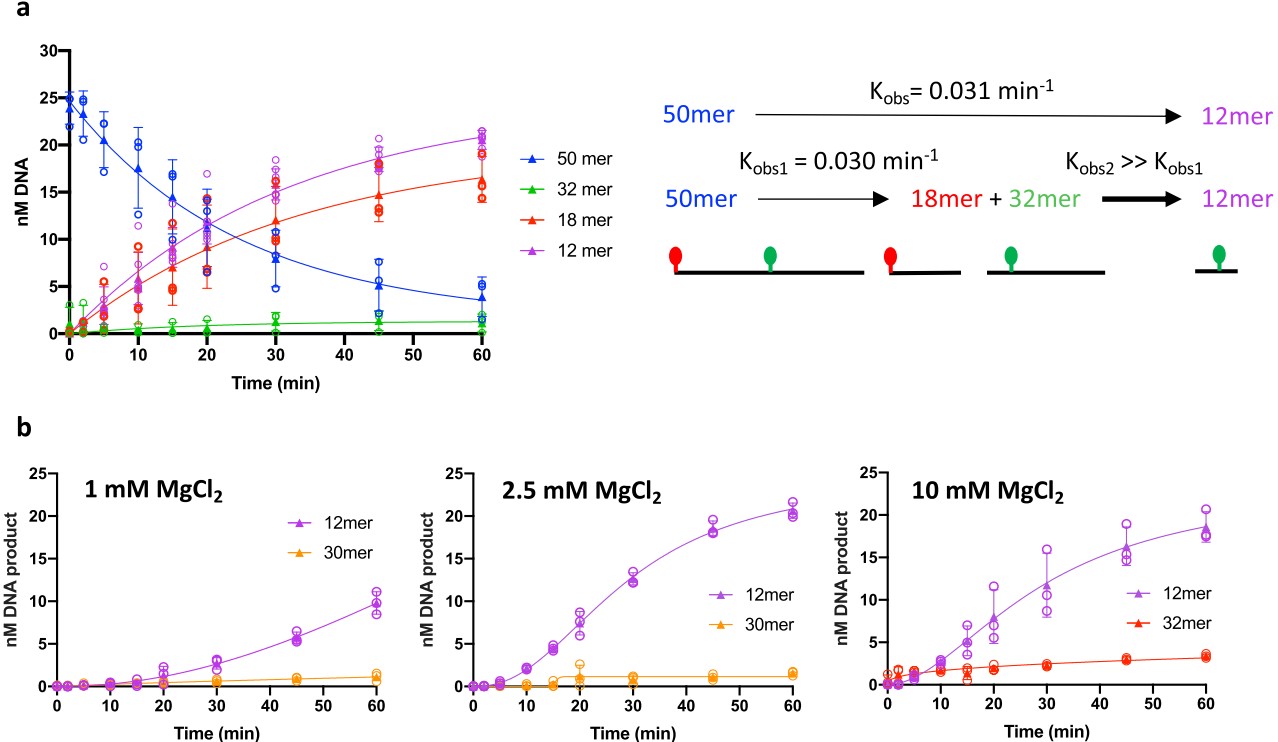

**Fig. 4 Kinetics of repair by drUvrABC. a** Time-course experiments following the changes in abundance of the different DNA fragments (50 mer in blue, 32 mer in green, 18 mer in red, and 12 mer in purple, illustrated to the right of the graph) as a function of time. The graph presents the mean (filled triangles) and standard deviation of six individual replicates shown as open circles. Reactions were performed at 37 °C using 25 nM F26-seq1 substrate, 1 μM drUvrA1, 0.5 μM drUvrB, and 2 μM drUvrC in the presence of 2.5 mM MgCl$_2$. Reactions were started with 2.5 mM ATP. The observed rate of product release, K$_{obs}$, corresponding to the rate of the dual-incision reaction, was determined to be 0.031 min$^{-1}$ by fitting the 12 mer data points to a single exponential model. The observed rate of 18 mer release, K$_{obs1}$, corresponding to the rate of the 5' incision reaction, was determined to be 0.030 min$^{-1}$ by fitting the 18 mer data points to a single exponential model. These findings suggest that the rate of the second incision reaction, K$_{obs2}$, must be much greater than the rate of the first incision. **b** Effects of MgCl$_2$ on the kinetics of drUvrABC dual-incision activity. Time-course experiments following the accumulation of the 12 mer product (purple) and the intermediate products, 30 mer (orange), resulting from 3' incision or 32 mer (red), resulting from 5' incision, as a function of time. The graphs present the mean (filled triangles) and standard deviation of three individual replicates shown as open circles. Reactions were performed at 37 °C using 25 nM F26-seq1 substrate, 1 μM drUvrA1, 0.5 μM drUvrB, and 2 μM drUvrC in the presence of 1 mM (left), 2.5 mM (middle), or 10 mM (right) MgCl$_2$.

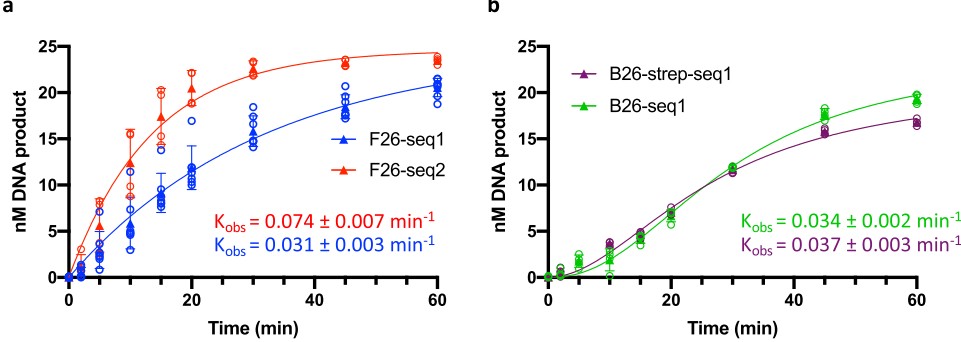

**Fig. 5 Substrate specificity of drUvrABC.** Kinetics of release of the 12 mer product by drUvrABC from either fluorescein-conjugated DNA substrates (**a**), F26-seq1 (blue) or F26-seq2 (red), or biotin-conjugated substrates (**b**), B26-seq1 (green) or B26-strep-seq1 (plum), detailed in Supplementary Table 3. The observed rates of product release, K$_{obs}$, corresponding to the rates of the dual-incision reaction, were determined for each substrate by fitting the data points to either a single exponential model (in **a**) or a sigmoidal model (in **b**). **a–b** Reactions were performed at 37 °C using 25 nM DNA substrate, 1 μM drUvrA1, 0.5 μM drUvrB, and 2 μM drUvrC in the presence of 2.5 mM MgCl$_2$. Reactions were started with 2.5 mM ATP. The graphs present the mean (filled triangles) and standard deviation of at least three individual replicates shown as open circles.

the binding affinity of drUvrA1 for the different DNA substrates used in the incision assay (Table 2 and Supplementary Fig. 6). drUvrA1 showed a tight binding (K$_d$ of 9 nM) to the FdT-containing substrates F26-seq1 and F26-seq2, with no difference in apparent affinity for these two substrates. In contrast, the affinity of drUvrA1 for B26-seq1 was much lower with an estimated Kd of 128 nM, i.e., almost 15-times higher than for FdT substrates and even higher than for a lesion-free DNA substrate

**Table 2 Kd values of drUvrA1 binding to different DNA substrates.**

| DNA substrate | Kd (nM) ± SE | Hill coeff. |
|---|---|---|
| F26-seq1 | 9 ± 0.7 | 1.34 |
| F26-seq2 | 9 ± 0.5 | 2.10 |
| B26-seq1 | 128 ± 15 | 0.97 |
| 5′-FAM-seq1 | 85 ± 12 | 0.92 |

(5′-FAM-seq1), which only bears a 5′-FAM label and no conjugated base in position 26.

MALDI-ToF mass spectrometry analyses of the products of the incision assays performed with F26-seq2 and B26-seq1 were carried out to determine the exact incision sites on these DNA substrates. As for F26-seq1, drUvrABC system was found to release, irrespective of the substrate, a 12 mer oligonucleotide with a phosphate at its 5′ end resulting from cleavage of the DNA 7 nucleotides upstream of the lesion and 4 nucleotides downstream of the lesion (Supplementary Tables 4, 5). The sites of incision are thus both sequence- and substrate-independent.

## Discussion

With this study, we report the development of a highly efficient in vitro NER system relying on the UvrA1, UvrB, and UvrC proteins from *D. radiodurans*. The *D. radiodurans* UvrA1 and B proteins typically display around 55% sequence identity with their homologs from model (*E. coli* or *Bacillus subtilis*) and pathogenic bacteria (*Pseudomonas aeruginosa*, *Mycobacterium tuberculosis*, *Helicobacter pylori*, or *Staphylococcus aureus*), while this level is a little lower (35–40%) for UvrC. Through improvements in the purification protocols of the three proteins and optimization of the reaction conditions (notably temperature, protein concentrations, salt, and metal-cation concentrations), we have established a robust repair system, capable of efficiently processing classical NER substrates incorporated into oligonucleotides, with near 80% completeness within 1 hour. The incision activity of drUvrABC is stable and can be followed for 2–3 hours at 37 °C, thereby providing us with a valuable tool to characterize this complex repair pathway and study its kinetics.

In the present study, we reveal that dual incision by drUvrABC requires all three Uvr proteins, ATP, and a divalent cation (Mg$^{2+}$ or Mn$^{2+}$). No incision activity was detected if one or more of these elements was left out of the reactions. The genome of *D. radiodurans* encodes for two UvrA variants, drUvrA1 and drUvrA2, which share similar structures and DNA-binding properties[35,37]. drUvrA1, however, cannot be substituted by drUvrA2 in the NER system, most likely because drUvrA2, which is missing a functional UvrB-binding domain, is unable to recruit UvrB and UvrC to the sites of DNA damage to form the pre-incision complex. In the absence of the two UvrA proteins, however, no incision activity was detected. In *E. coli*, in the absence of UvrA, UvrB and UvrC have been shown to locate DNA lesions[19,38] and incise the DNA either 5′ to existing single-strand cuts, but also to incise DNA close to the 5′ end of substrates in a damage-independent manner[17,20,27,39]. In suboptimal reaction conditions, and notably when Mn$^{2+}$ was used as the divalent cation instead of Mg$^{2+}$, we observed low levels of incision activity by drUvrB and drUvrC in the absence of drUvrA1, but this clearly appeared to be a spurious, non-specific damage-independent nuclease activity, since multiple bands were observed migrating just below the substrate. This was no longer observed in our optimal reaction conditions.

Interestingly, only low amounts of drUvrA1 were needed for efficient incision activity, whereas higher concentrations of

drUvrB and drUvrC were needed for optimal processing of the substrate, indicating that drUvrA1 is acting catalytically, as reported in earlier studies[12,13,40]. In the optimal reaction conditions, drUvrA1 binding to the FdT DNA substrates was very efficient, with a Kd close to 10 nM, as shown by the fluorescence-polarization measurements. Addition of drUvrA1 in a slight excess of the DNA concentration was thus sufficient to initiate the repair process. In contrast, the low binding affinity of drUvrB for drUvrA1 (as evidenced by the lack of complex formation by size-exclusion chromatography) may be a limiting step and may explain the need for higher concentrations of drUvrB and drUvrC for efficient repair.

Nucleotide binding and hydrolysis were also found to be essential for the dual incision activity of drUvrABC. ATP is known to be a key cofactor of bacterial NER and has been reported to play a role in regulating UvrA binding to DNA and its translocation along the DNA, but also DNA-damage recognition by UvrA and pre-incision complex formation, DNA opening and damage verification by UvrB, prior to incision by UvrC[19,20,26,27,35,37,41–49]. Both UvrA and UvrB possess nucleotide-binding domains. Each UvrA monomer possesses two ATP-binding sites, known as the proximal and distal sites[48], whereas UvrB has a single ATP binding site. In the present study we show that the drUvrABC functions optimally with a 1:1 ratio of ATP and Mg$^{2+}$, and that ADP or nonhydrolyzable analogs of ATP cannot substitute for ATP. Both ATP binding and hydrolysis by drUvrABC are therefore needed to allow the repair process to proceed, in agreement with studies of other bacterial NER systems using either nonhydrolyzable analogs or mutant forms of UvrA and UvrB that can no longer hydrolyze ATP[46,47]. ATP hydrolysis has notably been proposed to be essential for DNA-damage discrimination by UvrA and for the release of UvrA dimers from the DNA to allow formation of the pre-incision UvrB–DNA complex[46,48].

Divalent cations, and in particular magnesium, are well-known cofactors of nucleotide- and DNA-binding proteins. It was thus no surprise to find that the presence of a divalent cation was also a prerequisite for the dual incision activity of drUvrABC. Interestingly, only magnesium and manganese could support the incision activity, but addition of low concentrations of other metal cations, such as zinc, nickel, or iron, to reactions containing either magnesium, or manganese slightly increased the incision activity, suggesting that they may be accessory cofactors. A recent study has indeed shown that UvrC coordinates a [4Fe–4S] cluster[50]. Manganese in contrast could substitute quite efficiently for magnesium in this system. Magnesium and manganese have both been shown to be able to bind to the N-terminal GIY–YIG and the C-terminal RNase H endonuclease domains of *T. maritima* UvrC, although no experimental evidence was provided in these studies to demonstrate that manganese could also substitute for magnesium in the incision reaction containing *T. maritima* UvrC[23,24]. The high incision activity observed in the presence of manganese may be a particularity of the *D. radiodurans* NER system, since *D. radiodurans* is known to exhibit a high intra-cellular manganese concentration[51], which contributes to scavenging reactive oxygen radicals in conditions of oxidative stress[30,52].

In the presence of magnesium, we observed that the dual-incision activity was finely tuned by the concentration of the divalent ion in the reaction. Optimal activity was obtained at magnesium concentrations between 2.5 and 5 mM. At higher concentrations, such as 10 mM, which is the concentration typically used in most reported studies of bacterial NER activity[9,16,18,24], drUvrABC exhibited severely impaired activity. Low magnesium concentrations had an even more severe impact on the incision activity. Below 1 mM, the incision activity was

indeed barely detectable. Moreover, we noticed that the magnesium concentration differentially modulated the 3′ and 5′ incision reactions. High concentrations of magnesium affected the 3′ incision activity catalyzed by the N-terminal GIY–YIG domain more severely than the 5′ incision activity, while low-magnesium concentrations impacted both activities.

These experiments in which we varied the magnesium concentration also revealed that the drUvrABC system can perform the dual-incision reaction starting either with the 5′ cleavage site or with the 3′ cleavage site. Depending on the reaction conditions, we could detect either the 30 mer fragment resulting from 3′ cleavage or the 32 mer fragment resulting from 5′ cleavage. The first incision could thus occur on either side. Moreover, regardless of the side of the first cut, our kinetic analysis revealed that the second incision reaction follows very rapidly after the first incision indicating that the two reactions are sequential, but tightly coupled and thus quasi-simultaneous. The rate-limiting step is clearly the first incision, as reported in other bacterial NER systems[24,53,54]. However, in *E. coli*, incision occurs in a defined order with the 3′ incision first, followed by the 5′ incision[17]. The increased flexibility of the drUvrABC system may result from the experimental setup used in this assay, or could alternatively be a feature of *D. radiodurans* NER. Several DNA repair enzymes from *D. radiodurans* have indeed been shown to exhibit broader substrate specificity and additional or more robust catalytic activities than their counterparts from radio-sensitive model bacteria[55]. Functional plasticity may thus be a common trait of *D. radiodurans* DNA repair enzymes. The UvrD DNA helicase from *D. radiodurans* that is responsible for the release of the short 12 mer DNA fragment produced by drUvrABC has been shown, for example, to be a bipolar helicase capable of unwinding DNA duplexes in either the 5′–3′ or 3′–5′ directions, unlike *E. coli* UvrD which only unwinds duplexes in the 3′–5′ direction[56,57].

Analysis of the kinetics of repair of different DNA substrates by drUvrABC revealed two distinct reaction kinetics: a single exponential mode in which dual incision starts immediately and progresses during the reaction time course until completion, and a sigmoidal mode in which the incision reaction is initially very slow (lag phase) before increasing to reach similar reaction rates as observed in the exponential mode. We noticed that FdT-containing substrates followed the exponential mode, while BdT-containing substrates followed the sigmoidal mode. To better understand the mechanisms at play, we measured the binding affinity of drUvrA1 to these different DNA substrates. We found here again a clear distinction between these two substrates. drUvrA1 binds very tightly to FdT-containing substrates, but shows a 15-times lower affinity for the BdT-containing substrate. The lag phase at the start of the reactions with BdT–DNA could thus result from the slower binding of drUvrA1 to the modified base, which is the first step in the drUvrABC reaction. A similar lag phase was also observed on FdT substrates when lowering the temperature of the reaction, which may also slow the drUvrA1–DNA-binding kinetics. Interestingly, in these experiments we noticed a substantially higher reaction rate on the F26-seq2 substrate compared with the F26-seq1 substrate. These two substrates only differ in terms of DNA sequence. Their length, the nature and position of the lesion and the overall GC content of the oligonucleotides are the same. The binding affinities of drUvrA1 for these two substrates were also very similar. When looking more closely at the DNA sequence, we did notice, however, that locally in the vicinity of the FdT the DNA sequence of F26-seq2 was more AT-rich than the F26-seq1. We thus hypothesize that the energetic barrier for melting of the F26-seq2 substrate by the UvrB pre-incision complex may be lower than for the F26-seq1 substrate, which could explain the faster kinetics. Different sequence contexts have previously been reported to

affect incision by *E. coli* UvrABC[14]. It was also of interest to note that the bulkier substrate in which streptavidin was loaded onto BdT–DNA was not processed more efficiently than BdT–DNA alone, indicating that it is most likely local distortions of the DNA duplex that are recognized by drUvrABC and not the bulky adducts themselves.

Finally, MALDI-ToF mass spectrometry was performed to unambiguously characterize the sites of incision by drUvrABC. Unlike *E. coli* NER, which has been reported to release a 12 or 13 mer fragment resulting from incision at the 4th or 5th phosphodiester bond on the 3′ side and at the 8th phosphodiester bond on the 5′ side of the lesion[9,17,18], drUvrABC releases only a 12 mer fragment resulting from incision at the 4th phosphodiester bond on the 3′ side and at the 8th phosphodiester bond on the 5′ side of the lesion. The released 12 mer fragment bears a phosphate group at its 5′ extremity and thus corresponds to the following oligonucleotide: 5′p-NNNNNNNXNNN-3′, where N corresponds to any nucleotide and X to the damaged base. This configuration was observed, regardless of the nature of the substrate, suggesting that the sites of incision by drUvrABC are sequence- and damage-independent.

Developing a mesophilic NER system relying on the stable and robust drUvrABC proteins has thus provided us with a very valuable tool to investigate the mechanisms underlying bacterial NER, and will no doubt allow us and others in the future to decipher the precise roles of each Uvr protein in the recognition and repair of DNA lesions and better apprehend the substrate specificity of bacterial NER and its interplay with other repair pathways and the transcription machinery.

## Materials and methods

**Cloning, expression and purification of Uvr proteins.** *Deinococcus radiodurans* UvrA1 (DR_1771) and *UvrB* (DR_2275) genes were cloned into pProexHtB (EMBL). Based on sequence alignment, the first 59 amino acids of *D. radiodurans* UvrB (drUvrB) were removed and our full-length drUvrB construct thus starts at Met60 of the annotated DR_2275 sequence. The *UvrC* (DR_1354) gene was cloned into pET151d (Invitrogen). All constructs were expressed with cleavable N-terminal His-tags. drUvrA1 was expressed in BL21 (DE3) pLysS cells, while drUvrB and drUvrC were expressed in BL21 (DE3) cells. Cloning, expression and purification of *D. radiodurans* UvrA2 (DR_A0188) was described previously[35]. drUvrA1 expression was induced by 1 mM IPTG at 20 °C for 4 h in BL21 pLysS cells and cell pellets were resuspended in buffer A1 (20 mM Na-phosphate buffer, pH 8.0, 1 M NaCl, 5 mM β-mercaptoethanol (βME), and 1 mM MgCl₂) supplemented with protease inhibitors (Roche), DNase I (Roche) and lysozyme (Roche), and lysed by sonication. His-tagged drUvrA1 was initially purified on Ni-sepharose resin (GE Healthcare) and eluted with buffer A1 supplemented with 0.2 M imidazole. The cleavable N-terminal His-tag was removed by TEV digestion (1:20 w/w) overnight at 4 °C during the dialysis step into buffer A2 (20 mM Na-phosphate buffer, pH 8.0, 150 mM NaCl, and 1 mM tris(2-carboxyethyl)phosphine) (TCEP) to reduce the NaCl. After a Ni-IDA column (Macherey-Nagel) to separate the cleaved drUvrA1 from the His tag, the purified drUvrA1 was loaded on a 5 mL HiTrapQ column (GE Healthcare) and eluted with a NaCl gradient from 150 mM to 1 M NaCl. Finally, drUvrA1 was separated by size-exclusion chromatography on a SEC650 column (BioRad) in buffer A3 (50 mM Tris-HCl, pH 8.0, 150 mM NaCl, 1 mM TCEP and 10% glycerol). drUvrB expression was induced by 1 mM IPTG at 20 °C overnight. Cell pellets were resuspended in buffer B1 (50 mM Tris-HCl pH 8, 2 M NaCl, 10% sucrose and 2 mM MgCl₂) supplemented with protease inhibitors, DNase I and lysozyme, and were lysed by sonication. His-tagged drUvrB was initially purified on a 2 mL Ni-IDA column equilibrated in buffer B2 (20 mM Tris-HCl pH 8, 300 mM NaCl, 5 mM MgCl₂ and 2 mM βME) and eluted with buffer B2 supplemented with 0.25 M imidazole. The cleavable N-terminal His-tag was removed by TEV digestion (1:20 w/w) overnight at 4 °C during the dialysis step into buffer B3 (50 mM Tris-HCl pH 8, 150 mM NaCl, 1 mM MgCl₂, 1 mM TCEP and 5% glycerol) supplemented with 0.001% Brij35. The cleaved drUvrB was loaded on a 5 mL HiTrapQ column and eluted with a NaCl gradient from 150 mM to 1 M in buffer B3. Finally, drUvrB was separated by size exclusion chromatography on a SEC650 column in buffer B2 supplemented with 10% glycerol. drUvrC expression was induced by 1 mM IPTG at 20 °C overnight. Cell pellets were resuspended in buffer C1 (50 mM Tris-HCl pH 8, 2 M NaCl, 10% sucrose and 5 mM βME) supplemented with protease inhibitors, DNase I, lysozyme and S7 nuclease (Roche), and were lysed by sonication. His-tagged drUvrC was initially purified on a 2 mL Ni-IDA resin and eluted with Buffer C2 (50 mM Tris-HCl pH 8, 1 M NaCl, 10% glycerol and 2 mM βME) supplemented with 0.5 M imidazole. The fractions containing drUvrC were pooled, diluted to lower the NaCl concentration

to 300 mM and loaded on a 5 mL Heparin column (GE Healthcare) to eliminate DNA contamination. The protein was eluted with a NaCl gradient from 300 mM to 1 M in buffer C3 (50 mM Tris-HCl pH 8, 300 mM NaCl, 10% glycerol and 2 mM βME). The cleavable N-terminal His-tag was removed by TEV digestion (1:20 w/w) overnight at 4 °C. Finally, drUvrC was separated by size exclusion chromatography on a SEC650 column in buffer C4 (50 mM Tris-HCl pH 8, 500 mM NaCl, 10% glycerol, 5 mM MgCl$_2$ and 2 mM βME). The first batches of drUvrC were only partially active (Supplementary Fig. 2). Incomplete cleavage of the histidine tag resulted in a drUvrC protein with weak 5′ incision activity and no 3′ incision activity (batch 1 in Supplementary Fig. 2b), as previously reported for E. coli UvrC[17]. Tag cleavage by the TEV protease was most likely inhibited by the presence of nucleic acid contamination. This problem was resolved by performing a heparin affinity column prior to TEV cleavage. Moreover, we noticed that when using Ni-NTA resin, drUvrC stripped the nickel off the resin upon elution and the resulting protein was also partly inactive (batch 2 in Supplementary Fig. 2b). To avoid this, we used Ni-IDA resin in which the nickel ions are more tightly associated with the resin and were not removed during the chromatographic step. drUvrC possesses an iron-binding site in its N-terminal half close to the GIY-YIG endonuclease domain[50] and it is likely that the replacement of the iron by nickel interfered with the 3′ incision carried out by the GIY-YIG domain of drUvrC. All proteins were stored at −80 °C and were diluted in 50 mM Tris-HCl, pH 8.0, 150 mM NaCl, and 5% glycerol supplemented with freshly added 2 mM βME prior to use in the incision assay. This was critical to obtain high incision activity.

**DNA substrates and incision assay.** The sequences of the DNA substrates used in this study are given in Supplementary Table 2. All DNA oligonucleotides were ordered from MWG Biotech. Incision activity measurements were performed using duplexed 50 mer dsDNA oligonucleotides composed of a 5′-ATTO633 or FAM-labeled strand containing a conjugated thymine in position 26 annealed with an unlabeled complementary strand. The F26-seq1 substrate was composed of the 5′-ATTO633-F26-seq1 and Rev-seq1 strands, the F26-seq2 substrate was composed of the 5′-ATTO633-F26-seq2 and Rev-seq2 strands, and the B26-seq1 substrate was composed of the 5′-FAM-B26-seq1 and Rev-seq1 strands (Supplementary Table 3). The latter was used alone or in complex with streptavidin (B26-seq1-Strep). The DNA–streptavidin complex was formed by incubating the biotinylated B26-seq1 dsDNA (0.5 μM) with streptavidin (5 μM, Sigma) in 10 mM Tris-HCl, pH 8.0, 50 mM NaCl, and 0.5 mM EDTA at 25 °C for 30 min. Binding of streptavidin to the biotin-conjugated DNA was verified by native gel electrophoresis on a 10% TBE gel. Under these conditions, all the DNA was bound to streptavidin. Control duplexes containing an intact thymine in position 26 and a fluorophore-labelled 5′-end were also prepared, seq1-ATTO and seq1-FAM, composed respectively of either 5′-ATTO633-seq1 or 5′-FAM-seq1 strands annealed to Rev-seq1 (Supplementary Table 3). For the incision assay, a typical reaction involved incubation of 25 nM DNA substrate at 37 °C for 5 min with 1 μM drUvrA1, 0.5 μM drUvrB and 1 μM drUvrC in 50 mM Tris-HCl pH7.5, 50 mM KCl, 5 mM DTT and 2.5 mM MgCl$_2$ supplemented with 2 μM BSA, before initiating the incision reaction with the addition of 2.5 mM ATP. Reactions were stopped by addition of 10 μl stop buffer (2x TBE, 8 M urea, 0.025% bromophenol blue, and 0.1% SDS) to 10 μl reaction mix and subsequent heating of the samples to 95 °C for 5 min. Reactions were then analyzed on 20% TBE-8M urea polyacrylamide gels prerun at 5 W/gel in 1xTBE buffer. The gels were run for 35 min and the DNA bands were visualized and quantified on a Chemidoc MP imager (Bio-Rad) using the appropriate excitation light and detection filters for the green and red fluorophores, respectively. The respective amounts of each DNA fragment at a given time point were determined using ImageLab (Bio-Rad). Each experiment was performed at least three times and mean data points were fitted to appropriate models using GraphPad Prism 8. Kinetic data of product (12- or 18 mer fragments) release were fitted to either a single exponential model ($Y = A*(1-e^{-K_{obs}*X})$) or a sigmoidal model ($Y = A/[1 + (1/(K_{obs}*X)^h)]$) in which A is the amount of processed DNA (i.e., $Y_{max} - Y_{min}$), $K_{obs}$ is the observed rate of product release (in min$^{-1}$), and h is the Hill coefficient. Kinetic data of substrate (50 mer fragment) incision were fitted to a simple exponential decay model ($Y = A*(e^{-K_{obs}*X})$) using GraphPad Prism 8 in which A is the amount of processed DNA ($Y_{max} - Y_{min}$) and $K_{obs}$ is the observed rate of substrate incision.

**MALDI-ToF mass spectrometry.** Mass spectrometry measurements were performed using well-established protocols[58,59]. Incision reactions containing 2.5 pmol of DNA substrates in 100 μl reaction buffer were stopped at 60 min by heating samples to 95 °C for 5 min and were desalted and concentrated using C18 Ziptip pipette tips (Millipore). The tips were rinsed with water to remove the reaction buffer before the DNA was eluted with 10 μL 50/50 acetonitrile/H$_2$O. Then 1 μL of sample was added to 1 μL of 3-HPA matrix solution and the mix was spotted onto a MALDI-polished stainless target (Bruker) using the dried-droplet method. Mass spectra were obtained with a Microflex MALDI-ToF mass spectrometer (Bruker) operated in negative-ion mode and calibrated with reference oligonucleotides of known mass. As controls, mass spectra of the individual oligonucleotides composing the double-stranded substrates were also recorded.

**DNA-binding assays.** Equilibrium fluorescence anisotropy DNA-binding assays were performed on a Clariostar (BMG Labtech) microplate reader, fitted with polarization filters at room temperature. In all, 0–1 μM drUvrA1 (dimer) was titrated into 2 nM 5′-FAM-labeled substrates used for the incision assay (Supplementary Table 2) in binding buffer composed of 50 mM Tris, pH 7.5, 50 mM KCl, 5 mM DTT, 0.01% Tween 20, 2.5 mM MgCl$_2$, and 2.5 mM ATP supplemented with 0.1 mg/mL BSA. Reaction volumes were set to 50 μL. After subtracting the polarization values obtained for DNA alone, the mean data from at least three independent experiments were fitted to a standard binding equation ($Y = B_{max}*X^h/(K_d^h + X^h)$) assuming a single binding site with Hill slope (h) using GraphPad Prism 8, where $B_{max}$ is the difference between the anisotropy of completely bound and completely free oligo and $K_d$ is the equilibrium dissociation constant.

**Statistics and reproducibility.** As described above, incision activity assays and fluorescence-polarization experiments were performed multiple times. Presented data were derived from at least three independent experiments (distinct reaction mixes). Individual data points are shown in all presented line and bar graphs alongside the mean and standard-deviation values. The kinetics data were fitted to either a simple exponential or to a sigmoidal model, while the DNA-binding curves were fitted to a standard binding equation assuming a single binding site with Hill slope. All fits were prepared in GraphPad Prism 8 and the quality of the fits was assessed by checking the $R^2$ values. All fits showed $R^2$ values above 0.95.

**Reporting summary.** Further information on research design is available in the Nature Research Reporting Summary linked to this article.

## Data availability

The biochemical data (incision assay and DNA binding) that support the findings of this study[60] are available in figshare with the identifier https://doi.org/10.6084/m9.figshare.17212850. All other data produced during and/or analyzed during the current study are available from the corresponding author on reasonable request.

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

## Acknowledgements

IBS acknowledges integration into the Interdisciplinary Research Institute of Grenoble (IRIG, CEA). This project received funding from the IDEX University Grenoble Alpes, as part of the IRS 2017 call (CBS-IRS-2017-TIMMINS-RAVANAT). This work was supported by the Commissariat à l'énergie atomique et aux énergies renouvelables (CEA) through a radiobiology grant. This work used the platforms of the Grenoble Instruct-ERIC center (ISBG; UMS 3518 CNRS-CEA-UGA-EMBL) within the Grenoble Partnership for Structural Biology (PSB), supported by FRISBI (ANR-10-INBS-0005-02) and the GRAL and ARCANE labex, financed within the University Grenoble Alpes graduate school (Ecoles Universitaires de Recherche) CBH-EUR-GS (ANR-17-EURE-0003).

## Author contributions

Conceptualization: J.L.R and J.T.; Biochemical experiments and associated data analysis: A.S., S.D.B., J.L.R., and J.T.; MALDI-ToF analysis and data interpretation: C.S.-P. and D.G.; Writing and editing of the paper: A.S. and J.T. with input from all authors; Funding acquisition: J.L.R. and J.T.

## Competing interests

The authors declare no competing financial interests. Joanna Timmins is an Editorial Board Member for *Communications Biology*, but was not involved in the editorial

review of, nor the decision to publish this article. The remaining authors declare no competing interests.
