## [Peer Review File · Communications Biology]

Reviewers' comments:

Reviewer #1 (Remarks to the Author):

Deinococcus radiodurans contains both a UV-specific endonuclease encoded by the *uvrE* gene and a multi-subunit UvrABC endonuclease. Despite over 30 years of research in bacterial nucleotide excision repair, this is the first paper to purify and biochemically characterize the activity of the mesophilic UvrA, UvrB and UvrC proteins from *D. radiodurans*. Therefore, this initial biochemical characterization of the UvrABC system is an important contribution to the field. The authors show that the 5' incision is rate limiting in the dual incision process. They also show that the DrUvrABC system can work on SA bound to biotin, which is significant as previous studies suggested an upper limit of the ability to process protein-DNA crosslinks to 12-14 kDa. This limitation was apparently due to the inability to efficiently load UvrB at the damaged site for the *E. coli* proteins. For the most part the experiments are well described and executed. While it would be difficult for this group to recapitulate 30 years of work on the UvrABC system from other organisms in one paper, the following points would enhance the impact of this work.

1. For the general reader more information should be provided about UV resistance in *Deinococcus radiodurans*, it should be mentioned that in addition to two UvrA genes, that the organism also encodes a photoproduct endonuclease. Furthermore, a schematic showing how the two UvrA proteins differ, UvrA2 lacking the UvrB interaction domain for example, would be very helpful for some of the comparisons shown in the manuscript.
2. Figure 1 Panels A and B should be combined with Figure 2 and Panel 1C moved to the supplement as it shows that UvrC purity affects incision efficiency. How many times were the experiments in Figure 2 performed? The authors are encouraged to indicate the amount of product formed +/-S.D.
3. What is the nature of the large band migrating just below the 10mer marker in the red channel in both Figures 1 and 2?
4. Figure 3. The authors fail to realize that ATP must be bound to Mg²⁺ to be utilized by most proteins with ATPase activity. Thus, Figure 3B is of limited value since Mg²⁺ was only added to 2.5 mM. Is incision increased if an ATP regenerating system is supplied – it is possible that robust UvrA and UvrB ATPases might generate sufficient ADP levels that could cause a decrease in incision.
5. The lag in incision at 30C is quite interesting, could the authors please comment on this phenomena, what do they think is causing it?
6. The kinetic analysis in Figure 4 is excellent, but the Mass spec work-up of the products size should be introduced prior to this figure as the conclusions on the incision pattern can only be reached by knowing the nature of the incision sites. However, on line 279, I am not sure the authors can strongly state that the initial incision can occur on either the 5' or 3' side of the lesion. The data are not convincing.
7. Table 2 is valuable, but a comparison between UvrA1 and UvrA2 would be stronger. Please provide the binding isotherms in the supplement.
8. Unfortunately the authors try to link incision efficiency with UvrA binding affinity. This notion is just not correct and there is a large literature showing that UvrB-DNA complex formation is directly linked to incision efficiency.
9. The action mechanism of UvrABC suggests that UvrB and UvrC produce stoichiometric numbers of incisions, whereas UvrA should act catalytically and load multiple UvrB, thus the authors are encouraged to look at the optimal stoichiometry of the Uvr proteins to get maximal incision. They work at a huge protein excess over substrate.
10. Since UvrA2 lacks the UvrB interacting domain it should not be able to load UvrB at damaged sites. A direct test of this concept using EMSA analysis in which ATP and Mg are included in the gel and running buffer would be helpful.
11. Data provided in supplemental figure 1 is more compelling than data in Figure 1 and should be swapped out. Can one monomer of UvrA1 interact with another monomer of UvrA2? How long were the UvrA molecules premixed? Are the kinetics of the incision altered when UvrA2 is added to UvrA1? The authors only show one time point at 45 minutes. Perhaps it takes time for UvrA1 to load UvrB to get maximal incision by UvrB-UvrC which are in excess.

12. Supplemental Figure 3B is either miss labeled or the authors left out an important control of UvrC plus ATP and DNA.

13. The band positions in Supplemental Figure 4 at different Mg²⁺ concentrations is difficult to understand since the standards are overloaded; the 60 minute time points from 1, 2.5 and 10 mM Mg²⁺ should be run on the same gel to show the different length – it is impossible to tell if they are the lengths they indicate based on these gels.

14. The discussion on top of page 12 lines 380-389 is just not true. Over the years, the Goosen, Grossman, Sancar, Van Houten, and Zou laboratories have all done extensive kinetic analysis of the incision reactions on both plasmids and defined DNA substrates over various time courses. In fact, a great example can be found in Zou et al, JBC 273:12887-12892, 1998 in which heat shock proteins, DnaK, DnaJ, and GrpE were able to stabilize UvrA and provide as many as 10 cycles of loading of UvrB and subsequent incision. UvrA under these conditions acts catalytically. If there system is as robust as they say, can UvrA also act catalytically when used near substrate concentrations, but well above the K_d?

Reviewer #2 (Remarks to the Author):

The manuscript entitled "In vitro reconstitution of an efficient nucleotide excision repair system using 2 mesophilic enzymes from *Deinococcus radiodurans*" by Timmins and colleagues reports the invitro studies on NER system of this bacterium. They made recombinant UvrABC proteins with an utmost care to get high-quality proteins, reconstituted a multiprotein complex of UvrB and UvrC and both the variants of UvrA. Checked NER related activities in typical repair assay conditions, tested metal and ATP requirement of incision activities. Like several reports earlier, they could successfully constitute NER complex with UvrA1 protein in vitro albeit with some distinctions. This tool has helped authors to understand the kinetics of NER process in vitro. Apart from the first study with deinococcal proteins, the other new finding is the redundancy of UvrA2 in vitro NER function. What UvrA2 be doing in this bacterium would be worth answering through in vivo studies. Although, the results presented in this manuscript is clean with lesser ambiguity, the novelty of this study over the conventional research could not be ascertained. This reviewer finds it another data base addition in numerous studies on in vitro constitution of NER complex. The robust and efficient NER are 2 qualitative distinctions made in this study, to NER complex using Uvr system of this bacterium. In the opinion of this reviewer, these would be better supported if parallel studies with Uvr proteins from other bacterium that may not have such robust system has been carried out. This can be either by carrying out parallel study with other system or by discussing earlier published findings in a better quantifiable term.

Minor comments:

I find a lot of laboratory jargons described in results. A detailed procedure on polishing of proteins before actual experiments should be brought in method section. Also, a good amount of discussion components is mentioned in results. Overall, the discussion is too lengthy with repetition of some contents in introduction, results and discussion.

Reviewer #3 (Remarks to the Author):

The manuscript "In vitro reconstitution of an efficient nucleotide excision repair system using mesophilic enzymes from *Deinococcus radiodurans*" by A. Seck et al. presents kinetic studies of the nucleotide excision repair (NER) UvrABC system in *D. radiodurans*. Due to its high level of resistance to radiation induced DNA damage, *D. radiodurans* is an interesting organism for the study of DNA repair and specifically NER. Although in recent years, the eukaryotic (including the human) NER system that consists of the XP proteins (XPA-G) has been the major focus of research, the bacterial UvrABC system is still mechanistically highly interesting. In particular, potential differences between

extremely damage resistant *D. radiodurans* and other bacterial species may be important to better appreciate exact strategies of bacterial NER, which still remain incompletely understood. In their studies, the authors have probed the effects of different salt, ATP, and temperature conditions and have identified the exact incision product and the order of incisions for different bulky DNA damages by Uvr(AB)C from this organism. The data are of high scientific quality. The manuscript is well written and presents very interesting findings on the details of events in NER in *D. radiodurans*, which are also compared to (and display subtle differences to) previous data from other bacterial systems.

Nevertheless, I have some major issues with the manuscript, as listed below. For these reasons, I recommend the article by Seck et al. for publication in *Communications Biology* after major revisions.

(1) Introduction last paragraph, page 2/3: The authors should make clearer that the novelty of their incision assay lies in the clever combination of green and red fluorescence signals to specifically identify incision products. The incision assay per se is not novel and has been employed for many years to study details of DNA incisions by UvrABC from *E. coli* and thermophilic organisms (Zou and van Houten *EMBO J* 18(17), 4889-4901 (1999); Moolenaar et al. *JBC* 275(11), 8044-8050 (2000); Hoare et al. *Biochemistry* 39, 12252-12261 (2000); Verhoeven et al. *JBC* 275(7), 5120-5123 (2000); Wirth et al. *JBC* 291(36), 18932-18946 (2016)).

(2) Reconstitution of a functional NER system in vitro, Results page 6/7: While the optimization of drUvrC expression and purification offers interesting insight into functional requirements by the enzyme, the presence of a C-terminal his tag has already previously been reported to suppress 5' incision by *E. coli* UvrC (Verhoeven *JBC* 275(7), 5120-5123 (2000)). The authors should better set their findings for the *D. radiodurans* protein in the context of prior knowledge. Also, the presence of iron (in a 4Fe4S cluster) in the N-terminal Cys rich metal binding motif has recently been shown by the Barton laboratory (Silva, Grodick, and Barton, *JACS* (2020)) and this study may want to be cited in this context.

(3) Reconstitution of a functional NER system in vitro, Results page 7, second paragraph: The concentrations of UvrA, UvrB, and UvrC (1 μ M, 500 nM, 2 μ M, respectively) used in the assays are surprisingly high. In previous studies using enzymes from *E. coli* or thermophilic organisms, considerably lower concentrations have been employed: 2.5 nM- 30 nM for UvrA, 80 nM – 200 nM UvrB, 25 nM – 200 nM UvrC (Moolenaar et al. *JBC* 275(11), 8044-8050 (2000); Hoare et al. *Biochemistry* 39, 12252-12261 (2000); Verhoeven et al. *NAR* 30(11), 2492-2500 (2002); Wirth et al. *JBC* 291(36), 18932-18946 (2016)). These concentrations are much closer to the reported concentrations, at least in *E. coli* cells (\sim 10 nM UvrA, 100-200 nM UvrB, 10-50 nM UvrC). Although UvrA and UvrB concentrations are known to be upregulated in response to DNA damage, UvrC is not, so that the employed concentrations appear somewhat non-physiological. In fact, I believe that excessively high UvrA concentrations (of the order used in the studies reported here) have been reported to suppress NER incision activity likely due to nonspecific DNA coverage and subsequent blocking of loading of UvrB on the DNA. The authors may want to comment on their choice of protein concentrations in the assays. Were these concentrations also optimized in separate assays? Are the concentrations of UvrA/B/C in *D. radiodurans* known and much higher than in previously studied systems? This would seem relevant and interesting in the context of the high damage resistance of this organism.

(4) Reconstitution of a functional NER system in vitro, Results page 7 bottom: Nonspecific incision activity by UvrC in the presence of Mn²⁺ instead of Mg²⁺ is interesting, as is its disappearance upon addition of Fe³⁺. Did other metals also suppress the nonspecific incisions by UvrC? Is this related to the iron coordination in the iron sulfur cluster that has been reported to also affect DNA binding affinity by UvrC (Silva, Grodick, and Barton, *JACS* (2020))? In the text on page 7 it is stated that these nonspecific incisions occurred in the absence of UvrA, while in the figure caption (Supplemental Figure S3) UvrA was stated to be present in the reactions. Which is it (likely absence of UvrA since this is also stated in the Discussion on page 12)? Also: I wonder if the nonspecific incisions may be caused

by (artificially) high UvrC concentrations?

(5) Reconstitution of a functional NER system in vitro, Results page 8: Testing of different temperatures showed lag times in product formation for the lower temperatures (25°C). The authors may comment on the involvement of DNA binding affinity by the enzymes in this effect. Also when stating that at the higher temperature (42°C) “the reaction reached its maximal plateau after 30 minutes, indicating that the Uvr subunits may not be as stable at this temperature”, it would seem that the lower stability of the enzymes is not indicated by the fact that this maximum plateau is reached after 30 minutes, but that this plateau is considerably lower than at 37°C.

(6) Cleavage order and single-turnover repair kinetics, page 8: The first incision (or the conformational coordination required for this first incision) to be the rate limiting step has previously been reported for other bacterial organisms, as may want to be acknowledged. It has been shown that the 3' incision is the first incision in E coli as well as other previously investigated systems (as mentioned by the authors). The different and interchangeable order of the two DNA incisions observed here in these studies is interesting and may be related to either the different organism (with enhanced mechanistic flexibility for enhanced DNA repair in this extremely damage resistant organism) or due to different experimental conditions (salt or protein concentrations). Different orders of events appear to be seen for the different Mg²⁺ concentrations (see also next point). This as well as possible effects of the DNA sequence (that was also tested by the authors) may want to be addressed. Caron and Grossman also reported 5' before 3' incision “for CPD dimers in some DNA sequences” (J Cell Biochem 12, 359 (1988)).

(7) Cleavage order and single-turnover repair kinetics, page 8: I find the results at low (1 and 2.5 mM) and high (10 mM) Mg²⁺ interesting. At the low Mg²⁺ concentrations, some 3' incision without 5' is observed (supporting some suppression of 5' incision), while at 10 mM Mg²⁺ some 5' without 3' incision is clearly seen. The data at 10 mM Mg²⁺ seem to support the authors finding of the 5' incision being the first incision step, however, the data on which this conclusion was based were carried out at 2.5 mM Mg²⁺, at which condition here the opposite seems to be the case. Potentially different affinities for DNA at the different Mg²⁺ concentrations may want to be addressed in this context? Also: Gels in Suppl. Fig. S4 show the 30 mer (3' incision without 5') for 2.5 mM Mg while the plot in Figure 4 shows the 32-mer product (5' incision without 3') instead. Also: which Mg²⁺ concentration is physiologically relevant? What is the Mg²⁺ concentration in D. radiodurans? Is it lower than in other bacterial organisms, for which 10 mM Mg²⁺ have mostly been used in the reaction conditions? Are the differences (different order of incisions) observed in the studies presented here possibly related to the different Mg²⁺ concentration in the incubations (see also previous point)?

(8) Identification of the cleavage product, page 9/10: Fig. 5 shows mass spec data that unambiguously identify the incision product. The figure shows additional peaks in the m/z distributions for the initial substrate (prior to incisions) at ~8 kDa that may want to be commented on. Would it be possible to show the same m/z range for initial substrate and product?

(9) Substrate specificity, page 10/11: NER famously removes a large variety of lesion types. UvrABC incision positions for different types of lesions were reported here to be identical for different types of lesions. In fact, however, all of the investigated substrates contained bulky adducts (of different sizes). Previous studies have addressed Uvr(A)BC incisions for different types of lesions (Hoare et al. Biochemistry 39, 12252-12261 (2000)) and specifically for bulky lesions versus UV photoproducts (e.g. Wirth et al. JBC 291(36), 18932-18946 (2016)). In the Supplemental Tables S4 and S5 that provide mass spec data to identify incision products for fluorescein adducts and biotin adducts in the DNA substrate, respectively, “X” should be defined (also in Suppl. Table S1).

(10) Substrate specificity, page 10/11: DNA sequence dependence for UvrC incision activity and lag times for incisions of larger bulky lesions are reported. The possible slower conformational adjustment / need for protein reorientation that may be responsible for the delayed incisions of these larger

lesions may want to be (more clearly) addressed (supported by the anisotropy studies showing similar binding to the different substrates). Different incision activities for different types of lesions and different sequence context have previously been shown also for *E. coli* UvrABC (Verhoeven et al. NAR 30(11), 2492-2500 (2002)) and are consistent with the role of UvrA in opening the DNA duplex for loading of UvrB, which will be easier for DNA sequences that contain lower GC/AT ratios.

(11) Discussion, page 12: The authors claim that incubation times in previous incision studies could mostly not exceed 20 minutes due to instability of enzymes. Incubation times ≥ 30 min and up to several hours have in fact been applied for UvrABC from other bacterial systems (Hoare et al. Biochemistry 39, 12252-12261 (2000); Verhoeven et al. JBC 275(7), 5120-5123 (2000); Wirth et al. JBC 291(36), 18932-18946 (2016)).

(12) Discussion, page 12: The authors state that lack of DNA incisions by UvrBC in the absence of UvrA was surprising in their studies. In fact, it is not. In previously reported studies (by the van Houten and Tessmer laboratories (Zou and van Houten EMBO J 18(17), 4889-4901 (1999); Wirth et al. JBC 291(36), 18932-18946 (2016)), UvrBC in the absence of UvrA was only observed to accurately excise DNA lesions when an open DNA structure (DNA bubble) was introduced in the DNA substrate either at the lesion or closely upstream of the lesion. In those studies, the DNA bubble made the opening of the DNA by UvrA obsolete and allowed correct loading of UvrB(C) onto the DNA for lesion excision in the absence of UvrA. The cited study by the Kad laboratory indeed claims lesion recognition by UvrBC in the absence of UvrA and in the absence of loading sites in the DNA, however this study did not address DNA incisions by UvrC under these conditions.

(13) Discussion, page 12: The authors state "Interestingly, only low amounts of drUvrA1 were needed for efficient incision activity, whereas higher concentrations of drUvrB and drUvrC were needed for optimal processing of the substrate." I find it surprising then that such high UvrA concentrations (1 μ M) were used in the presented studies. Typically, much lower UvrA concentrations have been applied in these assays in the past (see above, point 3). Regarding the further discussed limiting low affinities between UvrA and UvrB, these are known to be modulated by ATP and DNA binding. Furthermore, binding affinities of UvrB and UvrC have been reported to be in range of 500 nM, at least for proteins from a different organism (Wirth et al. JBC 291(36), 18932-18946 (2016)), indeed partially supporting the authors' point of limiting affinities in the subcomplexes. However, previous studies have achieved DNA incisions by UvrABC and UvrBC at significantly lower concentrations of UvrA and UvrC.

(14) Discussion, page 13: ATP requirement for incisions by Uvr(AB)C are discussed. ATP hydrolysis requirements are, however, only shown for formation of the final incision product (Suppl. Fig. S2). Different requirements for ATP binding and hydrolysis have been previously shown for the 3' and 5' incisions in the *E. coli* system. In some of these studies, the different ATP requirements had been elegantly resolved by using DNA substrates, in which a nick in the DNA mimicked the first incision at the 3' side (e.g. Zou and van Houten EMBO J 18(17), 4889-4901 (1999); Moolenaar et al. JBC 275(11), 8044-8050 (2000)). In these studies, the second incision by UvrC, which in those systems was the 5' incision, did not depend on ATP. Other studies have addressed the requirement of ATP hydrolysis or ATP binding for DNA translocation and DNA opening versus DNA incisions by UvrC (Wirth et al. JBC 291(36), 18932-18946 (2016)).

Reviewers' comments

We wish to sincerely thank all three reviewers for their careful assessment of our manuscript and the constructive comments and suggestions they have made, which have no doubt improved the quality and clarity of our manuscript.

Below are our point-by-point answers to their comments.

Reviewer #1 (Remarks to the Author):

Deinococcus radiodurans contains both a UV-specific endonuclease encoded by the *UvsE* gene and a multi-subunit UvrABC endonuclease. Despite over 30 years of research in bacterial nucleotide excision repair, this is the first paper to purify and biochemically characterize the activity of the mesophilic UvrA, UvrB and UvrC proteins from *D. radiodurans*. Therefore, this initial biochemical characterization of the UvrABC system is an important contribution to the field. The authors show that the 5' incision is rate limiting in the dual incision process. They also show that the DrUvrABC system can work on SA bound to biotin, which is significant as previous studies suggested an upper limit of the ability to process protein-DNA crosslinks to 12-14 kDa. This limitation was apparently due to the inability to efficiently load UvrB at the damaged site for the *E. coli* proteins. For the most part the experiments are well described and executed. While it would be difficult for this group to recapitulate 30 years of work on the UvrABC system from other organisms in one paper, the following points would enhance the impact of this work.

1. For the general reader more information should be provided about UV resistance in *Deinococcus radiodurans*, it should be mentioned that in addition to two UvrA genes, that the organism also encodes a photoproduct endonuclease. Furthermore, a schematic showing how the two UvrA proteins differ, UvrA2 lacking the UvrB interaction domain for example, would be very helpful for some of the comparisons shown in the manuscript.

➤ As suggested by R1, we have added some information regarding the UV resistance of *D. radiodurans* to the introduction and have added a schematic diagram of UvrAs to the supplementary data (new Fig. S1).

New text pages 2 and 3: "*Deinococcus radiodurans* is a non-pathogenic, mesophilic bacterium, which displays an exceptional ability to withstand the lethal effects of DNA-damaging agents, including ionizing radiation and UV light. *D. radiodurans* can survive without loss of viability UV doses up to 500 J/m² (28, 29), a dose which is known to generate ~5,000 pyrimidine dimers per genome copy. A number of factors, including a high intracellular concentration of anti-oxidant metabolites, a well-protected proteome, an efficient DNA repair machinery and a high copy-number of its genome have been proposed to contribute to this robust radiation-resistant phenotype (30). The genome of *D. radiodurans* encodes for a complete NER pathway (31): UvrA1 (DR_1771), UvrB (DR_2275), UvrC (DR_1354) and UvrD (DR_1775), but also for a UV damage endonuclease, UvsE, that efficiently repairs UV-induced pyrimidine dimers (32). In a *uvrA1* knock out strain of *D. radiodurans*, UvsE can compensate in part for the absence of UvrA1 (33). In addition, a gene encoding for a second, class II UvrA protein (UvrA2; DR_A0188) can be found (31). Based on transcriptomics data, however, under normal growth conditions, UvrA2 is approximately ten times less abundant in *D. radiodurans* cells than its highly conserved counterpart, UvrA1 (34). The expression of all five *uvr* genes, but not the *UvsE* gene, is upregulated 3-5 fold following exposure to ionizing radiation (34). UvrA2 proteins are found in many bacteria living in harsh environments and show a high degree of sequence similarity to UvrA1, but are missing the proposed UvrB interaction domain (Supplementary Figure S1) (35). Despite this deletion, there is

evidence that UvrA2 may play a minor role in DNA repair and tolerance to DNA damaging agents, including UV, but it is unclear whether these UvrA variants are directly implicated in NER (33, 35). UvrA2 has also been proposed to take part in export of damaged oligonucleotides, a process that is known to occur in irradiated *D. radiodurans* (31).”

New Figure S1:

2. Figure 1 Panels A and B should be combined with Figure 2 and Panel 1C moved to the supplement as it shows that UvrC purity affects incision efficiency. How many times were the experiments in Figure 2 performed? The authors are encouraged to indicate the amount of product formed +/-S.D.

- As suggested by R1, we have combined Figures 1 and 2 (Figure 1 in revised manuscript) and moved the panel showing the effects of UvrC purity on the incision efficiency (Panel 2A, not 1C) to supplementary data (Figure S2). These experiments were performed several times, but only one representative gel is shown, because the results presented here are more qualitative than quantitative.

New Figure 1:

New Figure S2:

3. What is the nature of the large band migrating just below the 10mer marker in the red channel in both Figures 1 and 2?

- The large band corresponds to the loading dye running at the gel front that fluoresces in this red illumination channel – we observe this band even in the absence of DNA or protein. We have now specified this in the figure legend of Figure 1 and indicated it with an asterisk in panel D of Figure 1.

4. Figure 3. The authors fail to realize that ATP must be bound to Mg^{2+} to be utilized by most proteins with ATPase activity. Thus, Figure 3B is of limited value since Mg^{2+} was only added to 2.5 mM. Is incision increased if an ATP regenerating system is supplied – it is possible that robust UvrA and UvrB ATPases might generate sufficient ADP levels that could cause a decrease in incision.

- We are aware that ATP is very often utilized in complex with Mg^{2+} and the presented data confirms that ATP is indeed used in complex with Mg^{2+} by both ATPases (UvrA and UvrB) in our system. Having observed the effects of Mg^{2+} on the incision activity, we believed it was important to vary the ATP concentration also to define the optimal concentrations of each reagent. In fact, we observe that adding an excess of ATP actually causes a decrease in the incision efficiency, even though this should allow further cycles of ATPase. We also tried including an ATP regenerating system (based on phosphoenolpyruvate/pyruvate kinase), or adding more Mg^{2+} /ATP after 30 min of reaction, but these changes did not improve the system.

5. The lag in incision at 30C is quite interesting, could the authors please comment on this phenomena, what do they think is causing it?

- It is unclear what causes the lag at 30°C, but it is reminiscent of what is observed with poorer substrates, suggesting that the recognition by UvrA and/or the loading of UvrB and the formation of the pre-incision complex (including DNA melting) may be slower at this temperature.

6. The kinetic analysis in Figure 4 is excellent, but the Mass spec work-up of the products size should be introduced prior to this figure as the conclusions on the incision pattern can only be reached by

knowing the nature of the incision sites. However, on line 279, I am not sure the authors can strongly state that the initial incision can occur on either the 5' or 3' side of the lesion. The data are not convincing.

- The mass spec data has been moved up and is now presented before the kinetics analysis in the revised manuscript and the statement regarding the initial site of cleavage has been removed from this section.

7. Table 2 is valuable, but a comparison between UvrA1 and UvrA2 would be stronger. Please provide the binding isotherms in the supplement.

- Fluorescence polarization binding curves for UvrA1 have been added to the supplementary material as Fig. S6 (see below). We believe that a comparison of UvrA1 and UvrA2 is out of the scope of the present article, especially as DNA binding studies of UvrA2 have already been reported (Timmins et al, 2009) and in our NER assay, UvrA2 cannot substitute for UvrA1.

New Figure S6:

8. Unfortunately the authors try to link incision efficiency with UvrA binding affinity. This notion is just not correct and there is a large literature showing that UvrB-DNA complex formation is directly linked to incision efficiency.

- We propose that low incision efficiency may be due to poor UvrA1 binding in the case of Biotin-conjugated substrates, but we are aware that other steps could be affected as well, including the formation of the UvrB-DNA pre-incision complex. In fact, we propose that different efficiencies of UvrB-DNA complex formation could explain the differences in incision efficiency observed between the two FdT substrates seq1 and seq2 for which we do not see any difference in UvrA binding. This has been rephrased in the revised manuscript. New text page 15: "We thus hypothesize that the energetic barrier for melting of the F26-seq2 substrate by the UvrB pre-incision complex may be lower than for the F26-seq1 substrate, which could explain the faster kinetics. Different sequence contexts have previously been reported to affect incision by *E. coli* UvrABC (14)."

9. The action mechanism of UvrABC suggests that UvrB and UvrC produce stoichiometric numbers of incisions, whereas UvrA should act catalytically and load multiple UvrB, thus the authors are encouraged to look at the optimal stoichiometry of the Uvr proteins to get maximal incision. They work at a huge protein excess over substrate.

- We have performed numerous tests to find the optimal stoichiometry and concentrations of Uvr proteins to get maximal incision, and this is what we present as our optimal conditions: 1 μ M UvrA1, 0.5 μ M UvrB and 2 μ M UvrC. We can use lower amounts of UvrA1 (down to 0.1 μ M, still in excess of DNA) without any significant change in the incision efficiency (see Fig. S2), but at these lower concentrations, the assay appears to be less stable and the data are less reproducible – this is why we preferred to use 1 μ M in our optimal assay set-up. As shown in Table S3, we still had a robust activity on FdT substrates when using 10 times less protein (100 nM UvrA1, 50 nM UvrB and 200 nM UvrC), but at these concentrations, there was no detectable activity on Biotin-conjugated DNA (see Table S3). We are aware that in the optimal condition the proteins are in large excess over the substrate, but this is what is needed for single-turnover kinetics and to have a stable, functional assay.

10. Since UvrA2 lacks the UvrB interacting domain it should not be able to load UvrB at damaged sites. A direct test of this concept using EMSA analysis in which ATP and Mg are included in the gel and running buffer would be helpful.

- As stated above, we believe that further experiments focusing on the putative role and function of UvrA2 are out of the scope of the present study. They are of course of interest and will no doubt be the subject of future work.

11. Data provided in supplemental figure 1 is more compelling than data in Figure 1 and should be swapped out. Can one monomer of UvrA1 interact with another monomer of UvrA2? How long were the UvrA molecules premixed? Are the kinetics of the incision altered when UvrA2 is added to UvrA1? The authors only show one time point at 45 minutes. Perhaps it takes time for UvrA1 to load UvrB to get maximal incision by UvrB-UvrC which are in excess.

- We are unsure which figure panel R1 is referring to – we assume it is panel C in which different ratios of UvrA1/UvrA2 were tested – This panel has now been moved to Figure 1. We have never observed any complex formation between UvrA1 and UvrA2 (no co-elution from a gel filtration column for instance), but this is not surprising because in both cases the homodimers of UvrA1 and UvrA2 are very stable. In these experiments, the UvrA1 and UvrA2 proteins were pre-mixed for 10 min before starting the incision reaction. We only show the 45min timepoint in our graphs for reasons of clarity, but full kinetics were performed and we did not observe any change in the kinetics in the presence of UvrA2.

12. Supplemental Figure 3B is either miss labeled or the authors left out an important control of UvrC plus ATP and DNA.

- Fig. S3 (now Fig. S4) was indeed mislabeled and has now been corrected.

13. The band positions in Supplemental Figure 4 at different Mg²⁺ concentrations is difficult to understand since the standards are overloaded; the 60 minute time points from 1, 2.5 and 10 mM Mg²⁺ should be run on the same gel to show the different length – it is impossible to tell if they are the lengths they indicate based on these gels.

- R1 is right - the 30 and 32 mer bands cannot be differentiated easily based on their migration, but can be by their dye – the 32 mer band is only visible in the green channel whereas the 30 mer is doubly labelled in green and red and is thus visible in both the green and the red channels. In Figure S4 (now Fig. S5), a faint 30mer band can be seen at 1 and 2.5 mM MgCl₂, whereas the 32 mer band is seen at 10 mM MgCl₂ (this band is clearly missing in the red filter).

Supplementary Figure S5

14. The discussion on top of page 12 lines 380-389 is just not true. Over the years, the Goosen, Grossman, Sancar, Van Houten, and Zou laboratories have all done extensive kinetic analysis of the incision reactions on both plasmids and defined DNA substrates over various time courses. In fact, a great example can be found in Zou et al, JBC 273:12887-12892, 1998 in which heat shock proteins, DnaK, DnaJ, and GrpE were able to stabilize UvrA and provide as many as 10 cycles of loading of UvrB and subsequent incision. UvrA under these conditions acts catalytically. If there system is as robust as they say, can UvrA also act catalytically when used near substrate concentrations, but well above the K_d ?

- We apologize for this oversight regarding earlier kinetics studies. This paragraph has been removed. As mentioned above, when using drUvrA1 at lower concentrations (eg. 100 nM) with 25 nM substrate, we also expect it to be functioning catalytically. This has been added to the discussion of the revised manuscript on page 12: “Interestingly, only low amounts of drUvrA1 were needed for efficient incision activity, whereas higher concentrations of drUvrB and drUvrC were needed for optimal processing of the substrate, indicating that drUvrA1 is acting catalytically, as reported in earlier studies (12, 13, 42).”

Reviewer #2 (Remarks to the Author):

The manuscript entitled “In vitro reconstitution of an efficient nucleotide excision repair system using 2 mesophilic enzymes from *Deinococcus radiodurans*” by Timmins and colleagues reports the invitro studies on NER system of this bacterium. They made recombinant UvrABC proteins with an

utmost care to get high-quality proteins, reconstituted a multiprotein complex of UvrB and UvrC and both the variants of UvrA. Checked NER related activities in typical repair assay conditions, tested metal and ATP requirement of incision activities. Like several reports earlier, they could successfully constitute NER complex with UvrA1 protein in vitro albeit with some distinctions. This tool has helped authors to understand the kinetics of NER process in vitro. Apart from the first study with deinococcal proteins, the other new finding is the redundancy of UvrA2 in vitro NER function. What UvrA2 be doing in this bacterium would be worth answering through in vivo studies.

- Earlier studies have already tried to investigate the role of UvrA2 in vivo notably by knocking out either or both UvrAs in this organism (see paper by Tanaka et al, 2005). A description of this work has now been added to the introduction (page 2 and 3) in the part describing the UV resistance of *D. radiodurans*.

Although, the results presented in this manuscript is clean with lesser ambiguity, the novelty of this study over the conventional research could not be ascertained.

- In the revised manuscript, we have now highlighted the novel findings presented in this study at the end of the introduction (last paragraph).
New text page 3: "In the present study, we have established a robust incision assay relying on the activity of highly pure UvrA1, UvrB and UvrC from a single, mesophilic organism, the radiation resistant bacterium, *D. radiodurans* (31). In contrast to earlier work, this assay makes use of a doubly-labelled DNA substrate allowing us to specifically identify the different incision products. Moreover, this assay has enabled us to assess the possible involvement of UvrA2 in NER, to perform kinetics studies of the NER process, to probe the substrate specificity of *D. radiodurans* NER and to evaluate the role of divalent cations in the dual incision reaction. Finally, by combining this NER assay with MALDI-ToF mass spectrometry analyses, we unambiguously determined the sites of cleavage by UvrC."

This reviewer finds it another data base addition in numerous studies on in vitro constitution of NER complex. The robust and efficient NER are 2 qualitative distinctions made in this study, to NER complex using Uvr system of this bacterium. In the opinion of this reviewer, these would be better supported if parallel studies with Uvr proteins from other bacterium that may not have such robust system has been carried out. This can be either by carrying out parallel study with other system or by discussing earlier published findings in a better quantifiable term.

- Because the reaction conditions used in various earlier studies are different from those used in our assay (in terms of buffer, salt concentration, magnesium concentration, enzyme concentrations, temperature etc.) we feel that performing a reliable quantifiable comparison is simply not possible, which is why we have used qualitative distinctions.

Minor comments:

I find a lot of laboratory jargons described in results. A detailed procedure on polishing of proteins before actual experiments should be brought in method section. Also, a good amount of discussion components is mentioned in results. Overall, the discussion is too lengthy with repetition of some contents in introduction, results and discussion.

- We have moved some elements of laboratory jargon and details about the different batches of DrUvrC to the methods, and the gel showing the effect of UvrC purity on the incision efficiency has been moved to the Supplementary data (Figure S2B). We have also removed a

number of repetitions in the revised form. In particular, we have removed discussion components from the results section.

New text in Methods page 4 and 5: “The first batches of drUvrC were only partially active (Supplementary Figure S2). Incomplete cleavage of the histidine tag resulted in a drUvrC protein with weak 5' incision activity and no 3' incision activity (batch 1 in Supplementary Figure S2B), as previously reported for *E. coli* UvrC (17). Tag cleavage by the TEV protease was most likely inhibited by the presence of nucleic acid contamination. This problem was resolved by performing a heparin affinity column prior to TEV cleavage. Moreover, we noticed that when using Ni-NTA resin, drUvrC stripped the nickel off the resin upon elution and the resulting protein was also partly inactive (batch 2 in Supplementary Figure S2B). To avoid this, we used Ni-IDA resin in which the nickel ions are more tightly associated with the resin and were not removed during the chromatographic step. drUvrC possesses an iron-binding site in its N-terminal half close to the GIY-YIG endonuclease domain (36) and it is likely that the replacement of the iron by nickel interfered with the 3' incision carried out by the GIY-YIG domain of drUvrC. All proteins were stored at -80°C and were diluted in 50 mM Tris-HCl pH 8.0, 150 mM NaCl, 5% glycerol supplemented with freshly added 2 mM βME prior to use in the incision assay. This was critical to obtain high incision activity.”

Reviewer #3 (Remarks to the Author):

The manuscript “In vitro reconstitution of an efficient nucleotide excision repair system using mesophilic enzymes from *Deinococcus radiodurans*” by A. Seck et al. presents kinetic studies of the nucleotide excision repair (NER) UvrABC system in *D. radiodurans*. Due to its high level of resistance to radiation induced DNA damage, *D. radiodurans* is an interesting organism for the study of DNA repair and specifically NER. Although in recent years, the eukaryotic (including the human) NER system that consists of the XP proteins (XPA-G) has been the major focus of research, the bacterial UvrABC system is still mechanistically highly interesting. In particular, potential differences between extremely damage resistant *D. radiodurans* and other bacterial species may be important to better appreciate exact strategies of bacterial NER, which still remain incompletely understood. In their studies, the authors have probed the effects of different salt, ATP, and temperature conditions and have identified the exact incision product and the order of incisions for different bulky DNA damages by Uvr(AB)C from this organism. The data are of high scientific quality. The manuscript is well written and presents very interesting findings on the details of events in NER in *D. radiodurans*, which are also compared to (and display subtle differences to) previous data from other bacterial systems.

Nevertheless, I have some major issues with the manuscript, as listed below. For these reasons, I recommend the article by Seck et al. for publication in *Communications Biology* after major revisions.

(1) Introduction last paragraph, page 2/3: The authors should make clearer that the novelty of their incision assay lies in the clever combination of green and red fluorescence signals to specifically identify incision products. The incision assay per se is not novel and has been employed for many years to study details of DNA incisions by UvrABC from *E. coli* and thermophilic organisms (Zou and van Houten *EMBO J* 18(17), 4889-4901 (1999); Moolenaar et al. *JBC* 275(11), 8044-8050 (2000); Hoare et al. *Biochemistry* 39, 12252-12261 (2000); Verhoeven et al. *JBC* 275(7), 5120-5123 (2000); Wirth et al. *JBC* 291(36), 18932-18946 (2016)).

- This point has now been clarified at the end of the introduction and the additional references have been added when describing earlier NER assays (references 6, 9, 10, 12-27).

New text page 3: “In contrast to earlier work, this assay makes use of a doubly-labelled DNA substrate allowing us to specifically identify the different incision products.”

(2) Reconstitution of a functional NER system in vitro, Results page 6/7: While the optimization of drUvrC expression and purification offers interesting insight into functional requirements by the enzyme, the presence of a C-terminal his tag has already previously been reported to suppress 5' incision by *E. coli* UvrC (Verhoeven JBC 275(7), 5120-5123 (2000)). The authors should better set their findings for the *D. radiodurans* protein in the context of prior knowledge. Also, the presence of iron (in a 4Fe4S cluster) in the N-terminal Cys rich metal binding motif has recently been shown by the Barton laboratory (Silva, Grodick, and Barton, JACS (2020)) and this study may want to be cited in this context.

➤ These references have been added (references 17 and 36).

(3) Reconstitution of a functional NER system in vitro, Results page 7, second paragraph: The concentrations of UvrA, UvrB, and UvrC (1 μ M, 500 nM, 2 μ M, respectively) used in the assays are surprisingly high. In previous studies using enzymes from *E. coli* or thermophilic organisms, considerably lower concentrations have been employed: 2.5 nM- 30 nM for UvrA, 80 nM – 200 nM UvrB, 25 nM – 200 nM UvrC (Moolenaar et al. JBC 275(11), 8044-8050 (2000); Hoare et al. Biochemistry 39, 12252-12261 (2000); Verhoeven et al. NAR 30(11), 2492-2500 (2002); Wirth et al. JBC 291(36), 18932-18946 (2016)). These concentrations are much closer to the reported concentrations, at least in *E. coli* cells (~10 nM UvrA, 100-200 nM UvrB, 10-50 nM UvrC). Although UvrA and UvrB concentrations are known to be upregulated in response to DNA damage, UvrC is not, so that the employed concentrations appear somewhat non-physiological. In fact, I believe that excessively high UvrA concentrations (of the order used in the studies reported here) have been reported to suppress NER incision activity likely due to nonspecific DNA coverage and subsequent blocking of loading of UvrB on the DNA. The authors may want to comment on their choice of protein concentrations in the assays. Were these concentrations also optimized in separate assays? Are the concentrations of UvrA/B/C in *D. radiodurans* known and much higher than in previously studied systems? This would seem relevant and interesting in the context of the high damage resistance of this organism.

➤ See answer to point 9 in response to R1 regarding the choice of UvrABC concentrations and their effects on the dual incision activity. In *D. radiodurans*, transcriptomics studies have shown that all three Uvr proteins, UvrA1, UvrB, and UvrC, display similar relative abundance in non-irradiated cells, but are all up-regulated 3-5 fold following exposure to high doses of γ -irradiation (Liu et al, 2013). However, to our knowledge the concentrations of Uvr proteins in *D. radiodurans* are not known (or have never been reported). This information has been added to the introduction in the paragraph describing the UV resistance of *D. radiodurans*. New text on page 3: “Based on transcriptomics data, however, under normal growth conditions, UvrA2 is approximately ten times less abundant in *D. radiodurans* cells than its highly conserved counterpart, UvrA1 (34). The expression of all five *uvr* genes, but not the *uvrE* gene, is upregulated 3-5 fold following exposure to ionizing radiation (34).”

(4) Reconstitution of a functional NER system in vitro, Results page 7 bottom: Nonspecific incision activity by UvrC in the presence of Mn²⁺ instead of Mg²⁺ is interesting, as is its disappearance upon addition of Fe³⁺. Did other metals also suppress the nonspecific incisions by UvrC? Is this related to the iron coordination in the iron sulfur cluster that has been reported to also affect DNA binding affinity by UvrC (Silva, Grodick, and Barton, JACS (2020))? In the text on page 7 it is stated that these

nonspecific incisions occurred in the absence of UvrA, while in the figure caption (Supplemental Figure S3) UvrA was stated to be present in the reactions. Which is it (likely absence of UvrA since this is also stated in the Discussion on page 12)? Also: I wonder if the nonspecific incisions may be caused by (artificially) high UvrC concentrations?

- In our experiments with different metals, we observed incision activity in the presence of Mg, Mn or Fe only. Since the other metals showed no effects on the NER incision activity, they were not tested further to evaluate their potential effects on the non-specific incision by UvrB/UvrC (UvrA1 was indeed absent in this reaction, and the figure caption has now been corrected). As suggested by the reviewer, the observed effects of iron are likely related to the recently reported FeS cluster present in UvrC, which appears to be important in regulating the 3' incision reaction catalyzed by the N-terminal GIY-YIG domain of UvrC. Regarding the non-specific incision activity of UvrB/UvrC, we never observed this with Mg, and its extent was only very limited in the presence of Mn. The molecular mechanism underlying this difference remains to be determined, and may not be relevant *in vivo*, where indeed the levels of UvrC are likely much lower. There was indeed an error in the figure legend of Figure S3 (now Fig. S4) that has now been corrected.

(5) Reconstitution of a functional NER system *in vitro*, Results page 8: Testing of different temperatures showed lag times in product formation for the lower temperatures (25°C). The authors may comment on the involvement of DNA binding affinity by the enzymes in this effect. Also when stating that at the higher temperature (42°C) “the reaction reached its maximal plateau after 30 minutes, indicating that the Uvr subunits may not be as stable at this temperature”, it would seem that the lower stability of the enzymes is not indicated by the fact that this maximum plateau is reached after 30 minutes, but that this plateau is considerably lower than at 37°C.

- The effects of lowering the temperature on the binding of the enzymes to the DNA was in fact mentioned in the discussion when commenting on UvrA1 binding to the different substrates, since we observed similar lag-phases at the start of the reaction when using poor substrates (biotin-conjugated DNA) and when lowering the temperature to 30 and 25°C, indicating that they may both be due to slower UvrA1-DNA binding kinetics. We thank the reviewer for pointing out that indeed it is the low level of the plateau and not the fact that it is reached after 30 min that indicates a lower stability of the NER system at 42°C. This has now been corrected in the revised version.
New text on page 8: “Interestingly, no such lag phase was seen at 42°C, but the reaction reached a plateau after 30 minutes that was significantly lower than that observed at 37°C with only 20% of DNA incision, indicating that the Uvr subunits may not be very stable at this temperature (Figure 2D).”

(6) Cleavage order and single-turnover repair kinetics, page 8: The first incision (or the conformational coordination required for this first incision) to be the rate limiting step has previously been reported for other bacterial organisms, as may want to be acknowledged. It has been shown that the 3' incision is the first incision in *E. coli* as well as other previously investigated systems (as mentioned by the authors). The different and interchangeable order of the two DNA incisions observed here in these studies is interesting and may be related to either the different organism (with enhanced mechanistic flexibility for enhanced DNA repair in this extremely damage resistant organism) or due to different experimental conditions (salt or protein concentrations). Different orders of events appear to be seen for the different Mg²⁺ concentrations (see also next point). This as well as possible effects of the DNA sequence (that was also tested by the authors) may want to be addressed. Caron and Grossman also reported 5' before 3' incision “for CPD dimers in some DNA sequences” (*J Cell Biochem* 12, 359 (1988)).

- Several references have been added regarding the first incision being the rate-limiting step in other bacterial NER systems and the discussion of the observed flexibility in the order of incision by DrUvr proteins has been modified to take into account the reviewer's suggestions.

New text on page 14: "The rate-limiting step is clearly the first incision, as reported in other bacterial NER systems (24, 53, 54). However, in *E. coli*, incision occurs in a defined order with the 3' incision first, followed by the 5' incision (17). The increased flexibility of the drUvrABC system may result from the experimental set-up used in this assay, or could alternatively be a feature of *D. radiodurans* NER. Several DNA repair enzymes from *D. radiodurans* have indeed been shown to exhibit broader substrate specificity and additional or more robust catalytic activities than their counterparts from radio-sensitive model bacteria (55). Functional plasticity may thus be a common trait of *D. radiodurans* DNA repair enzymes."

(7) Cleavage order and single-turnover repair kinetics, page 8: I find the results at low (1 and 2.5 mM) and high (10 mM) Mg²⁺ interesting. At the low Mg²⁺ concentrations, some 3' incision without 5' is observed (supporting some suppression of 5' incision), while at 10 mM Mg²⁺ some 5' without 3' incision is clearly seen. The data at 10 mM Mg²⁺ seem to support the authors finding of the 5' incision being the first incision step, however, the data on which this conclusion was based were carried out at 2.5 mM Mg²⁺, at which condition here the opposite seems to be the case. Potentially different affinities for DNA at the different Mg²⁺ concentrations may want to be addressed in this context? Also: Gels in Suppl. Fig. S4 show the 30 mer (3' incision without 5') for 2.5 mM Mg while the plot in Figure 4 shows the 32-mer product (5' incision without 3') instead. Also: which Mg²⁺ concentration is physiologically relevant? What is the Mg²⁺ concentration in *D. radiodurans*? Is it lower than in other bacterial organisms, for which 10 mM Mg²⁺ have mostly been used in the reaction conditions? Are the differences (different order of incisions) observed in the studies presented here possibly related to the different Mg²⁺ concentration in the incubations (see also previous point)?

- We agree with the reviewer that the results obtained at low and high Mg²⁺ concentrations are interesting and indicate that both 5' and 3' incision are partly impaired at low Mg²⁺, whereas the 3' incision is clearly more affected at high Mg²⁺.

New text clarifying this point has been added on pages 10 of the results and on pages 13-14 of the discussion: (i) page 10: "Interestingly, we observed that at 1 mM Mg²⁺, the 12 mer fragment production dropped significantly, indicating that at this Mg²⁺ concentration, the rate of incision is severely reduced. The very low amounts of intermediate products (in this case, the 30 mer fragment) seen to accumulate under these conditions suggest that the cleavage reactions are nonetheless still quasi-simultaneous as at 2.5 mM MgCl₂. This indicates that both the 5' and 3' incision reactions are impaired at low magnesium concentration. In contrast, at 10 mM Mg²⁺, we observed a marked accumulation of the intermediate 32 mer fragment caused by a reduced 3' incision activity. Mg²⁺ thus plays a very critical role in fine-tuning the dual incision activity of drUvrABC." (ii) pages 13-14: "High concentrations of magnesium affected the 3' incision activity catalyzed by the N-terminal GIY-YIG domain more severely than the 5' incision activity, while low magnesium concentrations impacted both activities. These experiments in which we varied the magnesium concentration also revealed that the drUvrABC system can perform the dual incision reaction starting either with the 5' cleavage site or with the 3' cleavage site. Depending on the reaction conditions, we could detect either the 30 mer fragment resulting from 3' cleavage or the 32 mer fragment resulting from 5' cleavage. The first incision site could thus occur on either side."

In this system, magnesium plays multiple roles – it is a co-factor of ATP required for the ATPase activities of UvrA1 and UvrB, it is a co-factor of UvrC known to be involved in the catalytic

activities of both endonuclease domains, and thirdly it is likely to modulate the binding of each of the Uvr proteins on the DNA substrate. It thus seems difficult to postulate on the possible mechanisms underlying the observed differences in incision obtained at low and high Mg^{2+} . We agree that different DNA binding affinities could be an explanation, but without additional insight we feel it is too early to propose a possible mechanism. Regarding the discrepancy between the graph presented in Figure 4B and the gel shown in Fig. S4 (now Fig. S5), we thank the reviewer for pointing this out. This was a mistake and has now been corrected. At 2.5 mM Mg^{2+} , the dual incision activity is quasi-simultaneous and only very low amounts of intermediate fragments are seen. We often observed the 32 mer fragment as shown in the data presented in Fig. 4A, but in some instances we also observed the 30 mer fragment as illustrated in Fig. S5 and now also in Fig. 4B. In all cases, the levels of these bands are very low since the dual incision is quasi-simultaneous at 2.5 mM $MgCl_2$. To our knowledge, nothing is known regarding the physiological magnesium concentration in *D. radiodurans* and how it relates to that in other bacteria.

(8) Identification of the cleavage product, page 9/10: Fig. 5 shows mass spec data that unambiguously identify the incision product. The figure shows additional peaks in the m/z distributions for the initial substrate (prior to incisions) at ~8 kDa that may want to be commented on. Would it be possible to show the same m/z range for initial substrate and product?

- Additional peaks were indeed observed at a m/z around 8000 and correspond to the doubly charged oligonucleotides $(M-2H)^{2-}$, as indicated in Panel A of Figure 3 (new numbering). This is normal to see both singly and doubly charged forms of the same oligonucleotides. The m/z of the doubly charged form is half of the m/z of the singly charged forms. This has been clarified in the figure legend of Fig. 3 (new numbering). We prefer not to use the same m/z range for the mass spectra of starting and cleaved oligonucleotides since there is a large difference in mass between the two and this would make the observation of the 12, 20 and 18 mer products very difficult.

(9) Substrate specificity, page 10/11: NER famously removes a large variety of lesion types. UvrABC incision positions for different types of lesions were reported here to be identical for different types of lesions. In fact, however, all of the investigated substrates contained bulky adducts (of different sizes). Previous studies have addressed Uvr(A)BC incisions for different types of lesions (Hoare et al. *Biochemistry* 39, 12252-12261 (2000)) and specifically for bulky lesions versus UV photoproducts (e.g. Wirth et al. *JBC* 291(36), 18932-18946 (2016)). In the Supplemental Tables S4 and S5 that provide mass spec data to identify incision products for fluorescein adducts and biotin adducts in the DNA substrate, respectively, "X" should be defined (also in Suppl. Table S1).

- X has now been defined in the footnotes of Tables S4 and S5.

(10) Substrate specificity, page 10/11: DNA sequence dependence for UvrC incision activity and lag times for incisions of larger bulky lesions are reported. The possible slower conformational adjustment / need for protein reorientation that may be responsible for the delayed incisions of these larger lesions may want to be (more clearly) addressed (supported by the anisotropy studies showing similar binding to the different substrates). Different incision activities for different types of lesions and different sequence context have previously been shown also for *E. coli* UvrABC (Verhoeven et al. *NAR* 30(11), 2492-2500 (2002)) and are consistent with the role of UvrA in opening the DNA duplex for loading of UvrB, which will be easier for DNA sequences that contain lower GC/AT ratios.

- The relation between poorer binding of substrate by UvrA1 (as seen by anisotropy studies) and longer lag times has been clarified in the revised manuscript as well as the influence of different GC/AT content on the ability of UvrA1 and UvrB to open the DNA duplex to form the pre-incision complex. The suggested reference has also been added as reference 14.

(11) Discussion, page 12: The authors claim that incubation times in previous incision studies could mostly not exceed 20 minutes due to instability of enzymes. Incubation times ≥ 30 min and up to several hours have in fact been applied for UvrABC from other bacterial systems (Hoare et al. *Biochemistry* 39, 12252-12261 (2000); Verhoeven et al. *JBC* 275(7), 5120-5123 (2000); Wirth et al. *JBC* 291(36), 18932-18946 (2016)).

- We apologize for this oversight. This paragraph has been removed. The suggested references are referenced elsewhere in the manuscript (references 21, 17 and 27).

(12) Discussion, page 12: The authors state that lack of DNA incisions by UvrBC in the absence of UvrA was surprising in their studies. In fact, it is not. In previously reported studies (by the van Houten and Tessmer laboratories (Zou and van Houten *EMBO J* 18(17), 4889-4901 (1999); Wirth et al. *JBC* 291(36), 18932-18946 (2016)), UvrBC in the absence of UvrA was only observed to accurately excise DNA lesions when an open DNA structure (DNA bubble) was introduced in the DNA substrate either at the lesion or closely upstream of the lesion. In those studies, the DNA bubble made the opening of the DNA by UvrA obsolete and allowed correct loading of UvrB(C) onto the DNA for lesion excision in the absence of UvrA. The cited study by the Kad laboratory indeed claims lesion recognition by UvrBC in the absence of UvrA and in the absence of loading sites in the DNA, however this study did not address DNA incisions by UvrC under these conditions.

- In this section, the reviewer may have misunderstood our message. We were not surprised to see that drUvrA1 was needed for DNA incision by drUvrBC, but wished to stress that unlike some reported NER systems, we did not see any non-specific cleavage by UvrBC alone (except when replacing Mg^{2+} with Mn^{2+}) and that all three Uvr proteins were needed for efficient DNA incision. UvrA1 plays a major role in localizing the sites of damage and loading UvrB and UvrC onto these sites, notably by favoring the formation of a DNA bubble as indicated by the reviewer. We have clarified this section to avoid any misunderstanding, have corrected the sentence regarding the single-molecule study by the Kad laboratory and have added the proposed references.
New text on page 12: "*E. coli* UvrB and UvrC have been shown to be able to locate DNA lesions in the absence of UvrA (19, 41) and non-specific incision activity by UvrB and UvrC has previously been reported in other bacterial NER systems (17, 27)."

(13) Discussion, page 12: The authors state "Interestingly, only low amounts of drUvrA1 were needed for efficient incision activity, whereas higher concentrations of drUvrB and drUvrC were needed for optimal processing of the substrate." I find it surprising then that such high UvrA concentrations (1 μ M) were used in the presented studies. Typically, much lower UvrA concentrations have been applied in these assays in the past (see above, point 3). Regarding the further discussed limiting low affinities between UvrA and UvrB, these are known to be modulated by ATP and DNA binding. Furthermore, binding affinities of UvrB and UvrC have been reported to be in range of 500 nM, at least for proteins from a different organism (Wirth et al. *JBC* 291(36), 18932-18946 (2016)), indeed partially supporting the authors' point of limiting affinities in the subcomplexes. However, previous studies have achieved DNA incisions by UvrABC and UvrBC at significantly lower concentrations of UvrA and UvrC.

- As mentioned previously, efficient incision could be achieved with lower UvrA1 concentrations (in the 50-100 nM range), but under these conditions, the assay was more variable, perhaps because UvrA1 is less stable in the assay conditions at low concentration, so we chose to define the optimal conditions as those that yielded the strongest, but also the most reproducible incision activity. As for the affinity between the Uvr proteins, we definitely observe some differences with proteins from other organisms. As mentioned in the manuscript, we cannot isolate a stable UvrA1/UvrB complex using the proteins from *D. radiodurans*, indicating that the K_d is most likely $> 1-10 \mu\text{M}$, but in contrast we can easily purify a very stable UvrB/UvrC complex (undoubtedly submicromolar affinity), which has been the focus of another study (article in preparation).

(14) Discussion, page 13: ATP requirement for incisions by Uvr(AB)C are discussed. ATP hydrolysis requirements are, however, only shown for formation of the final incision product (Suppl. Fig. S2). Different requirements for ATP binding and hydrolysis have been previously shown for the 3' and 5' incisions in the *E. coli* system. In some of these studies, the different ATP requirements had been elegantly resolved by using DNA substrates, in which a nick in the DNA mimicked the first incision at the 3' side (e.g. Zou and van Houten EMBO J 18(17), 4889-4901 (1999); Moolenaar et al. JBC 275(11), 8044-8050 (2000)). In these studies, the second incision by UvrC, which in those systems was the 5' incision, did not depend on ATP. Other studies have addressed the requirement of ATP hydrolysis or ATP binding for DNA translocation and DNA opening versus DNA incisions by UvrC (Wirth et al. JBC 291(36), 18932-18946 (2016)).

- We have added these references to our discussion of the different roles of ATP binding and hydrolysis in the NER activities.
New text on page 12: "ATP is known to be a key co-factor of bacterial NER and has been reported to play a role in regulating UvrA binding to DNA and its translocation along the DNA, but also DNA damage recognition by UvrA and pre-incision complex formation, DNA opening and damage verification by UvrB, prior to incision by UvrC (19, 20, 26, 27, 35, 40, 43-50)."

REVIEWERS' COMMENTS:

Reviewer #1 (Remarks to the Author):

The authors have adequately responded to the concerns raised in the first review. The manuscript and study now read better and will have a larger impact on the field.

The authors should cite this paper in the paragraph in the discussion (lines 418-431) regarding the role of ATP in the activity of the UvrABC system:

Barnett JT, Kad NM. Understanding the coupling between DNA damage detection and UvrA's ATPase using bulk and single molecule kinetics. *FASEB J.* 2019 Jan;33(1):763-769. PMID: 30020831; PMCID: PMC6355085.

Reviewer #3 (Remarks to the Author):

Most of my comments have been adequately addressed in the revised manuscript. I have only very few points remaining that should be easily modified in the manuscript:

(1) In my opinion the most novel and remarkable (simple but clever) innovation in the incision assay developed by the authors is the use of two fluorophores to distinguish between 3' and 5' incision by UvrABC. I think this should be integrated in the abstract, in the second last sentence that addresses the newly developed assay.

(2) In the revised Supplemental Figure S2, the red and green channel gels are shown. At this point the concept of this assay (the interpretation of the different incision products based on their specific red only, green only, or red and green fluorescence) has not been introduced yet, so this is likely confusing for readers. I would suggest to only show the green channel gel with the specific incision product here, since the red channel gel is not really important in this context. Alternatively, if the authors want to refer back to this figure later in the manuscript and discuss the incision products seen in the respective fluorescence channels, then the assay may want to be briefly explained in the figure caption.

(3) In Results line 238 Figure 1B should be 1C.

(4) In Supplemental Figure S4B, the non-specific incision product for Mn²⁺ is still present in the green channel gel (red arrow in top right red channel gel).

(5) Discussion lines 402/3: this may be a misunderstanding but to my mind non-specific incision implies lesion independent incision. This is not the case in the studies cited here (references 17 and 27) where incisions were lesion specific but required a loading site (an unpaired DNA region) in the absence of UvrA. The Goosen lab has, however, reported additional 5' incisions by UvrC (5' from existing single strand cuts, Moolenaar et al. 1998 *JBC* 273) and UvrBC has been shown to load on (and excise) DNA damage close to the 5' end of DNA substrate in the absence of UvrA (Moolenaar et al. 2000 The role of ATP binding and hydrolysis by UvrB during NER, *JBC* 275(11)).

(6) My previous point (9): The conserved incision sites for the different NER target lesions studied here may not be so surprising (in spite of their varying degree of bulkiness) since they all represent bulky adducts attached to a DNA base - in contrast to UV photoproducts that involve crosslinking of bases and as a consequence stronger DNA distortions.

Point-by-point response to reviewer's comments:

REVIEWERS' COMMENTS:

Reviewer #1 (Remarks to the Author):

The authors have adequately responded to the concerns raised in the first review. The manuscript and study now read better and will have a larger impact on the field.

The authors should cite this paper in the paragraph in the discussion (lines 418-431) regarding the role of ATP in the activity of the UvrABC system:

Barnett JT, Kad NM. Understanding the coupling between DNA damage detection and UvrA's ATPase using bulk and single molecule kinetics. FASEB J. 2019 Jan;33(1):763-769. PMID: 30020831; PMCID: PMC6355085.

- Reference has been added to the discussion, now listed as reference 49.

Reviewer #3 (Remarks to the Author):

Most of my comments have been adequately addressed in the revised manuscript. I have only very few points remaining that should be easily modified in the manuscript:

(1) In my opinion the most novel and remarkable (simple but clever) innovation in the incision assay developed by the authors is the use of two fluorophores to distinguish between 3' and 5' incision by UvrABC. I think this should be integrated in the abstract, in the second last sentence that addresses the newly developed assay.

- As suggested by R3, we have added this point to the abstract. "This newly developed assay relying on the use of an original, doubly-labelled DNA substrate has allowed us to probe the kinetics of repair on different DNA substrates and to determine the order and precise sites of incisions on the 5' and 3' sides of the lesion."

(2) In the revised Supplemental Figure S2, the red and green channel gels are shown. At this point the concept of this assay (the interpretation of the different incision products based on their specific red only, green only, or red and green fluorescence) has not been introduced yet, so this is likely confusing for readers. I would suggest to only show the green channel gel with the specific incision product here, since the red channel gel is not really important in this context. Alternatively, if the authors want to refer back to this figure later in the manuscript and discuss the incision products seen in the respective fluorescence channels, then the assay may want to be briefly explained in the figure caption.

- Supplementary Figure S2 has been modified accordingly. The red-boxed gel has been removed from Panel B and a brief sentence has been added to the figure legend to explain the colored (red and green) boxes in panels A and B. Panel A: "The gel was

visualized with the red filter to detect ATTO633-labelled bands.” Panel B: “The gels were visualized with the green filter to detect fluorescein-labelled bands.”

(3) In Results line 238 Figure 1B should be 1C.

- We thank the reviewer for pointing out this mistake. We have corrected this in the revised manuscript.

(4) In Supplemental Figure S4B, the non-specific incision product for Mn²⁺ is still present in the green channel gel (red arrow in top right red channel gel).

- A second red arrow has been added to the green channel gel to avoid any confusion.

(5) Discussion lines 402/3: this may be a misunderstanding but to my mind non-specific incision implies lesion independent incision. This is not the case in the studies cited here (references 17 and 27) where incisions were lesion specific but required a loading site (an unpaired DNA region) in the absence of UvrA. The Goosen lab has, however, reported additional 5' incisions by UvrC (5' from existing single strand cuts, Moolenaar et al. 1998 JBC 273) and UvrBC has been shown to load on (and excise) DNA damage close to the 5' end of DNA substrate in the absence of UvrA (Moolenaar et al. 2000 The role of ATP binding and hydrolysis by UvrB during NER, JBC 275(11)).

- This sentence has been edited to avoid any misunderstanding regarding specific- or non-specific incision activity, and the two suggested references have been added as suggested by R3 to illustrate respectively lesion-specific incision activity occurring at “non-specific” sites and spurious cleavage occurring in the absence of lesions. The modified text is: “In *E. coli*, in the absence of UvrA, UvrB and UvrC have been shown to locate DNA lesions (19, 41) and incise the DNA either 5' to existing single-strand cuts, but also to incise DNA close to the 5' end of substrates in a damage-independent manner (17, 20, 27, 42).” Ref 39 corresponds to Moolenaar, JBC, 1998.

(6) My previous point (9): The conserved incision sites for the different NER target lesions studied here may not be so surprising (in spite of their varying degree of bulkiness) since they all represent bulky adducts attached to a DNA base - in contrast to UV photoproducts that involve crosslinking of bases and as a consequence stronger DNA distortions.

- This is correct and we hope to use our assay to probe the sites of incision on a broader range of DNA lesions including UV photoproducts in the near future.